# On Connecting Deep Trigonometric Networks with Deep Gaussian Processes: Covariance, Expressivity, and Neural Tangent Kernel

## Abstract

Deep Gaussian Process (DGP) as a model prior in Bayesian learning intuitively exploits the expressive power of function composition. DGPs also offer diverse modeling capabilities, but inference is challenging because marginalization in latent function space is not tractable. With Bochner's theorem, DGP with squared exponential kernel can be viewed as a deep trigonometric network consisting of the random feature layers, sine and cosine activation units, and random weight layers. In the wide limit with a bottleneck, we show that the weight space view yields the same effective covariance functions which were obtained previously in function space. Also, varying the prior distributions over network parameters is equivalent to employing different kernels. As such, DGPs can be translated into the deep bottlenecked trigonometric networks, with which the exact maximum a posteriori estimate can be obtained. Interestingly, the network representation enables the study of DGP's neural tangent kernel, which may also reveal the mean of the intractable predictive distribution. Statistically, unlike the shallow networks, deep networks of finite width have covariance deviating from the limiting kernel, and the inner and outer widths may play different roles in feature learning. Numerical simulations are presented to support our findings.

## 1 Introduction

Nearly a decade has passed since the proposal of Deep Gaussian Process (DGP) (Damianou & Lawrence, 2013) which, along with principled uncertainty estimation inherited from Gaussian Process (GP) (Rasmussen & Williams, 2006), aimed to exploit the compositional structure like Deep Neural Network (DNN) for superior expressivity and feature learning. Unfortunately, adopting DGP in application remains difficult due to costly computation and challenging optimization (Dutordoir et al., 2021). In the Bayesian setting, computation of exact posterior is impossible because one must marginalize multiple latent functions within the hierarchy. Numerous approximate Bayesian inference schemes, see e.g. (Bui et al., 2016; Salimbeni & Deisenroth, 2017; Ustyuzhaninov et al., 2020), have been proposed. Because of the intractability of inference, seemingly basic questions, e.g. the expressivity of DGP, remain unanswered. Analytic methods, even only for maximum a posteriori (MAP), would allow further insights.

One particular approximate DGP inference stands out among others as it does not rely on imposing inducing points on latent functions and makes strong connection with DNN. Cutajar et al. (2017) utilized the concept of expanding the squared exponential (SE) kernels in terms of Gaussian random features and sine/cosine activation (Rahimi & Recht, 2008), which allows one to translate a GP with SE kernel into a shallow but infinitely wide trigonometric network. Then DGP, as a cascade of GPs, is a random deep bottlenecked network (Agrawal et al., 2020), i.e. the activation layers have infinite units but latent output layers are of finite dimension. The bottlenecks ensure the heavy-tailed statistics (Pleiss & Cunningham, 2021) pertaining to DGPs (Duvenaud et al., 2014; Lu et al., 2020), unlike the DNNs without bottlenecks are converged into GP (Lee et al., 2018; Matthews et al., 2018). To pursue MAP of DGP in this context, we shall show that varying prior over the weight parameters translates to different kernel compositions for DGPs (Lu et al., 2020). Thus, we can apply gradient descent to the squared loss minus the log of prior over weights for

obtaining a MAP estimate. More interestingly, the MAP solutions shall be closely related to those obtained from the neural tangent kernel (NTK) regression (Jacot et al., 2018; Arora et al., 2019).

Therefore, the deep bottlenecked networks position us to understand the true expressive power of DGPs, including whether simply stacking GPs is better than the tricks of kernel composition (Duvenaud et al., 2013; Wilson et al., 2016; Sun et al., 2018) and activation design (Pearce et al., 2020). Nevertheless, DGPs offer appealing flexibility such as multi-fidelity modeling (Kennedy & O'Hagan, 2000; Cutajar et al., 2019; Lu & Shafto, 2021a) and can be regarded as a Bayesian deep kernel learning (Wilson et al., 2016; Ober et al., 2021; Lu & Shafto, 2021b). Another critical issue is the general lack of feature learning for kernel based models like GP and DGP. Kernel functions are fixed, not depending on training data whereas the features learned in DNNs are result of back propagating the training error. We shall analyze the finite-width kernels of the random deep bottlenecked networks, the results of which suggest that the learning with a finite-width Bayesian deep network is similar with GP learning but with random kernels (Benton et al., 2019).

In this paper, we pursue analytical results and investigate the two-layer wide bottlenecked trigonometric network, a proxy of two-layer DGP with SE kernels, and make four main contributions. (i) Covariance: we show the equivalence between the two models as the bottlenecked random networks in the wide limit yield the same exact covariance (Lu et al., 2020). (ii) Expressivity: we show shallow trig networks can approximate a GP with spectral mixture kernel (Wilson & Adams, 2013) if the features are samples from mixture of Gaussians. In addition, marginal prior distribution (Yaida, 2020; Zavatone-Veth & Pehlevan, 2021) of a shallow trig net can be non-Gaussian if an embedding phase shift network is incorporated. (iii) NTK: translating DGPs to the deep trigonometric network representation allows us to derive a closed form NTK for the corresponding DGPs. The expectation is that kernel regression using NTK shall correspond to the exact MAP solution of DGP. (iv) Finite-width effects: We define a kernel estimator for a finite network by marginalizing the random weight parameters. The kernel estimator is then a function of the random features. Mean of the estimator only coincides with the exact DGP kernel in the wide limit, which signifies the difference with the shallow network (Yu et al., 2016).

The paper has the following organization. A background for the trigonometric networks, deep Gaussian processes, and the random feature expansion of kernels is introduced in Sec. 2. In Sec. 3, covariance of shallow trig networks with different parameter distributions and its non-Gaussian function distribution are discussed. The derivation of effective kernels for deep trigonometric networks with various parameter distributions is given in Sec. 4. As the connection between deep trigonometric network and DGP is built, Sec. 5 is devoted to the derivation of neural tangent kernel. Considering the reality for neural networks, Sec. 6 formulates the framework for calculating the correction to covariance as a result of the finite width. Numerical simulations are presented in Sec. 7. The context of literature in which the present work should be placed can be found in Sec. 8, and a conclusion in Sec. 9 is provided.

## 2 Background

Consider a parametric function $f_{\mathbf{W}}(\mathbf{x})$ which maps input $\mathbf{x} \in \mathbb{R}^D$ to real number. In Bayesian settings, given the data $\{\mathbf{x}_i, y_i\}_{i=1:N}$ denoted by $\mathcal{D}$, the goal is to obtain the predictive distribution,

$$p(y_*|\mathbf{x}_*, \mathcal{D}) = \int d\mathbf{W} \ p(y_*|f_{\mathbf{W}}(\mathbf{x}_*))p(\mathbf{W}|\mathcal{D}) \,,$$

for an unseen input $\mathbf{x}_*$. A simple likelihood is Gaussian density, $p(y|f_{\mathbf{W}}(\mathbf{x})) = \mathcal{N}(y|f_{\mathbf{W}}(\mathbf{x}), \sigma_n^2)$. The posterior is obtained through Bayes rule, $p(\mathbf{W}|\mathcal{D}) \propto p(\mathbf{W}) \prod p(y_i|f_{\mathbf{W}}(\mathbf{x}_i))$, with the normalization constant being the evidence or marginal likelihood. In most cases for Bayesian deep neural networks, the marginalization over the parameters $\mathbf{W}$ is not tractable, and one may seek the maximum a posteriori (MAP) solution. Namely, $p(y_*|f_{\overline{\mathbf{W}}}(\mathbf{x}_*))$ becomes the predictive solution with

$$\overline{\mathbf{W}} = \operatorname{argmin}\left[ -\log p(\mathbf{W}) - \sum \log p(y_i|f_{\mathbf{W}}(\mathbf{x}_i)) \right] \,.$$

To understand the translation between the weight and function representations, we shall analytically investigate i) the marginal function prior for a single input and ii) the covariance $\int d\mathbf{W} f_{\mathbf{W}}(\mathbf{x}) f_{\mathbf{W}}(\mathbf{y}) p(\mathbf{W})$ in weight representation for comparing with the covariance obtained in function representation.

As the basis for theoretical findings in this paper, we outline three prior theoretical results: marginal prior distribution for deep linear neural network (Zavatone-Veth & Pehlevan, 2021), exact covariance of two-layer DGP with squared exponential kernel (Lu et al., 2020), and the random feature expansion of squared exponential kernel (Rahimi & Recht, 2008).

## 2.1 Random neural networks

Neural networks are a class of parametric models in which one can regard the function output as the outcome of propagating the input through a computational graph consisting of multiple layers of linear and nonlinear mappings. For example, a shallow network can have the following form,

$$f(\mathbf{x}) = \mathbf{w}\Phi(\Omega\mathbf{x}) \,, \tag{1}$$

where the input $\mathbf{x} \in \mathbb{R}^D$ is sequentially propagated through the feature layer (producing preactivation from multiplying $\Omega \in \mathbb{R}^{n \times d}$ with input), activation units (element-wise nonlinear mapping $\Phi(\cdot)$), and weight layer (multiplying $\mathbf{w} \in \mathbb{R}^{1 \times n}$). A deep neural network has similar structure. For instance,

$$f(\mathbf{x}) = \mathbf{w}_2\Phi\big(\Omega_2\mathbf{W}_1\Phi(\Omega_1\mathbf{x})\big) \,, \tag{2}$$

where the matrices in feature layers have $\Omega_1 \in \mathbb{R}^{n_1 \times D}$ and $\Omega_2 \in \mathbb{R}^{n_2 \times H}$, and in weight layers $\mathbf{W}_1 \in \mathbb{R}^{H \times n_1}$ and $\mathbf{w}_2 \in \mathbb{R}^{1 \times n_2}$. The integer $H$ represents the width of latent layer output in deep networks.

The inductive bias associated with neural networks is connected to the prior distributions from which the random parameters in the computational graph are sampled. How well a model can generalize in Bayesian learning is critically related to its inductive bias (Wilson & Izmailov, 2020). While it is usually difficult to describe the inductive bias of neural networks quantitatively, some special cases do permit analytic investigation. Zavatone-Veth & Pehlevan (2021) analytically investigated the marginal distribution over the output of deep linear and ReLu networks. The following remark is about a particular shallow linear network.

**Remark 1.** *Consider the linear network $f(\mathbf{x}) = \mathbf{W}\Omega\mathbf{x}$, a special case of Eq. (1) with $\Phi$ being identity mapping, and the entries in the random matrices $\Omega \in \mathbb{R}^{2 \times D}$ and $\mathbf{W} \in \mathbb{R}^{1 \times 2}$ are independent normal, i.e. $\Omega_{ij} \sim \mathcal{N}(0, \sigma_1^2)$ and $\mathbf{W}_{ij} \sim \mathcal{N}(0, \sigma_2^2)$. Then, the marginal distribution over the output is a Laplace distribution $p(f(\mathbf{x})) = \exp(-|f(\mathbf{x})|/\kappa)/2\kappa$ with $\kappa := \sigma_1\sigma_2|\mathbf{x}|$. The heavy-tailed character is consistent with the findings in (Vladimirova et al., 2019).*

*Proof.* It is easy to observe that the latent output $\mathbf{h} = \Omega\mathbf{x}$ has independent components $h_i \sim \mathcal{N}(0, \sigma_1^2|\mathbf{x}|^2)$. Similarly, conditional on $\mathbf{h}$, the output has $f|\mathbf{h} \sim \mathcal{N}(0, \sigma_2^2|\mathbf{h}|^2)$. To obtain the marginal distribution $p(f) := \mathbb{E}_{\mathbf{W},\Omega}[p(f|\mathbf{W},\Omega)] = \mathbb{E}_{\mathbf{h}}[p(f|\mathbf{h})]$, one can integrate out $\mathbf{h}$ during the Fourier transformation and then apply the inverse transform (Zavatone-Veth & Pehlevan, 2021). Namely, in this particular case with hidden dimension $n = \dim(\mathbf{h}) = 2$, we can get,

$$p(f) = \int \frac{dq}{2\pi}d\mathbf{h} \; e^{iqf}\tilde{p}(q|\mathbf{h})p(\mathbf{h}) = \int \frac{dq}{2\pi} \frac{e^{iqf}}{1 + \sigma_1^2\sigma_2^2|\mathbf{x}|^2q^2} = \frac{e^{-|f|/\kappa}}{2\kappa} \,. \tag{3}$$

In deriving the above, we have used the fact that the Fourier transformation of $p(f|\mathbf{h})$ is $\tilde{p}(q|\mathbf{h}) = \exp(-\frac{1}{2}q^2\sigma_2^2|\mathbf{h}|^2)$ and the residue theorem is applied to complete the last equality. □

As the outputs of neural network are not independent given the shared parameters, another perspective of studying the inductive bias is to investigate the distribution over the function values, i.e. $p(f(\mathbf{x}_1), f(\mathbf{x}_2), \cdots, f(\mathbf{x}_N))$, indexed by the set of inputs. This is a more challenging task than the above marginal distribution over the function at single input. Fortunately, the central limit theorem applies when the number of activation units becomes infinity, the multivariate distribution converges to Gaussian, and the limiting statistics only depends on the mean $\mathbb{E}[f(\mathbf{x})]$ and covariance $\mathbb{E}[f(\mathbf{x})f(\mathbf{y})]$. Closed form covariance functions can be derived for shallow networks with sigmoidal and ReLu activations (Williams, 1997; Cho & Saul, 2009), but the same techniques do not seem to carry to the deeper networks. As for the deep networks of finite width, various techniques from statistical physics (Dyer & Gur-Ari, 2019; Yaida, 2020; Roberts et al., 2021) have been employed to compute the corrections.

## 2.2 Gaussian process and deep Gaussian process

In parallel, Gaussian Processes (Rasmussen & Williams, 2006) (GPs) directly model the set of function values with a Gaussian, $p(f(\mathbf{x}_{1:N})|\theta) = \mathcal{N}(\mu(\mathbf{x}_{1:N}), \Sigma(\mathbf{x}_{1:N}, \mathbf{x}_{1:N}))$, with $\theta$ being the hyper-parameters in the mean function $\mu$ and covariance matrix $\Sigma$, which fully specify the model. Being Gaussian allows analytic marginalization, which leads to the defining property of the mean function $\mathbb{E}[f(\mathbf{x}_i)] = \mu(\mathbf{x}_i)$ and the covariance function,

$$\Sigma_{ij} = \mathbb{E}\{[f(\mathbf{x}_i) - \mu(\mathbf{x}_i)][f(\mathbf{x}_j) - \mu(\mathbf{x}_j)]\} = k(\mathbf{x}_i, \mathbf{x}_j) \, ,$$

where $k$ is a predetermined kernel function, e.g. squared exponential function. In addition, a closed form for the marginal likelihood $p(\mathbf{y}|\mathbf{X}, \theta) = \mathbb{E}_{f \sim \mathcal{N}(\mu, \Sigma)}[p(\mathbf{y}|f(\mathbf{X}))]$ can be obtained if a Gaussian likelihood is adopted, with which the optimal hyper-parameters is determined. Conditional on the prior observations, the responses $\mathbf{y}_*$ at a set of inputs $\mathbf{X}_*$ then follows another normal distribution $\mathcal{N}(\mathbf{y}_*|\mu_*, \Sigma_*)$ with conditional mean,

$$\mu_* = \Sigma(\mathbf{X}_*, \mathbf{X})[\Sigma(\mathbf{X}) + \sigma_s^2 I]^{-1} \mathbf{y} \, , \tag{4}$$

and conditional covariance,

$$\Sigma_* = \Sigma(\mathbf{X}_*) - \Sigma(\mathbf{X}_*, \mathbf{X})[\Sigma(\mathbf{X}) + \sigma_s^2 I]^{-1} \Sigma(\mathbf{X}, \mathbf{X}_*) \, , \tag{5}$$

where we take the prior mean to be zero, $\mu = 0$, for easing the notation, and hyper-parameter $\sigma_s^2$ denoting the noise variance connecting $f$ to the observations.

Among many extensions of GPs for enhancing expressivity, e.g. warped GP in (Snelson et al., 2004), Deep Gaussian Processes (DGPs) (Damianou & Lawrence, 2013) are a general hierarchical composition of GPs. The compositional structure enhances its expressive power, e.g. a GP with SE kernel can not fit a step function well but a DGP can. Consider for simplicity a two-layer function $f(\mathbf{x}) = f_2(\mathbf{f}_1(\mathbf{x}))$ where the input $\mathbf{x} \in \mathbb{R}^D$ is mapped to the hidden output $\mathbf{h} = \mathbf{f}_1(\mathbf{x}) \in \mathbb{R}^H$ and then to a real output $f_2(\mathbf{h})$. The hidden layer with finite $H$ is referred to as the bottleneck in (Agrawal et al., 2020; Aitchison, 2020). DGP is defined by the joint density $p(f_2(\mathbf{f}_1(\mathbf{x}_{1:N})))$,

$$\mathcal{N}(f_2(\mathbf{h}_{1:N})|0, \Sigma_2(\mathbf{H})) \prod_{i=1}^{H} \mathcal{N}(h_i(\mathbf{x}_{1:N})|0, \Sigma_1(\mathbf{X})) \, ,$$

where subscripts in covariance matrices remind us that the covariance functions in different layers can be different. The hidden output $\mathbf{H}$ is a data matrix consisting of vector-valued hidden functions $\mathbf{h}(\mathbf{x}_{1:N})$, entering as input to second GP. In Bayesian inference, the marginalization of the hidden random variables $\mathbf{h}$ is not tractable, which leads to various approximation schemes including variational inference (Salimbeni & Deisenroth, 2017; Salimbeni et al., 2019; Yu et al., 2019; Ustyuzhaninov et al., 2020; Ober & Aitchison, 2021) and expectation propagation (Bui et al., 2016).

One advantage of modeling with the function space view, such as GP, is that we can augment the model by imposing constraint on the function through inducing points (Titsias, 2009; Titsias & Lawrence, 2010), i.e. the random function has to pass through a set of points, $f(\mathbf{z}_{1:M}) = u_{1:M}$, in the absence of noise. Those points can be treated as additional hyper-parameters to be optimized (empirical Bayes), or can be treated as random variable so that one has to infer their distribution in a full Bayes setting. In the context of DGP, these inducing points can serve as hidden function's support in variational inference (Salimbeni & Deisenroth, 2017), or they can be interpreted as the low fidelity observations in multi-fidelity regression problems (Kennedy & O'Hagan, 2000; Cutajar et al., 2019). However, it becomes less straightforward to incorporate these inducing points into the deep neural networks from a random weight space view (Ober & Aitchison, 2021).

An alternative scheme for inference with DGP models is to view DGP as a GP at the level of the marginal prior, i.e. the hidden function $\mathbf{f}_1$ being marginalized out from the joint, which is similar to the partially collapsed inference in Gibbs sampling (Park & Van Dyk, 2009) and deep kernel learning (Wang et al., 2020). The idea was motivated by the observation that the covariance of the marginal prior distribution over the array of function values taken at inputs $\mathbf{X}$,

$$p(\mathbf{f}|\mathbf{X}) = \int d\mathbf{F}_1 \, p(\mathbf{f}_2|\mathbf{F}_1) p(\mathbf{F}_1|\mathbf{X}) \, , \tag{6}$$

can be computed analytically (Lu et al., 2020). As such, an approximating distribution $q(\mathbf{f}|\mathbf{X}) = \mathcal{N}(0, \Sigma_{\text{eff}})$ with the matched covariance $[\Sigma_{\text{eff}}]_{ij} = \mathbb{E}[f_2(\mathbf{f}_1(\mathbf{x}_i))f_2(\mathbf{f}_1(\mathbf{x}_j))]$ can be plugged into the standard GP inference pipeline. The compositional hierarchy incorporates all scales from layers into the effective kernels, e.g. $k_{\text{eff}} = \sigma_2^2 \{1 + 2\frac{\sigma_1^2}{\ell_2^2}[1 - \exp(-\frac{d^2(\mathbf{x}_i, \mathbf{x}_j)}{2\ell_1^2})]\}^{-\frac{1}{2}}$ for 2-layer DGP with SE kernels in both zero-mean GPs, and the multi-scale character enables capturing complex patterns in some time series data (Lu & Shafto, 2021b). Moreover, the model augmentation incorporating latent function supports as additional hyper-parameters was shown to have better generalization (Lu & Shafto, 2021b). The closed form kernel for the 2-layer DGP with learnable latent function support is in the following lemma. The proof can be found in (Lu & Shafto, 2021a).

**Lemma 1.** *Consider the two-layer DGP, $f(\mathbf{x}) = f_2(\mathbf{f}_1(\mathbf{x}))$, where the latent functions, $\mathbf{f}_1 : \mathbb{R}^D \mapsto \mathbb{R}^H$ being a vector-valued GP and $f_2 : \mathbb{R}^H \mapsto \mathbb{R}$ being a GP with SE kernel. The latent function is conditioned on the support, $\mathbf{f}_1(\mathbf{z}_{1:M}) = \mathbf{u}_{1:M}$. The covariance has the following closed form (Lu & Shafto, 2021a),*

$$\mathbb{E}_{\mathbf{f}_1} \mathbb{E}_{f_2|\mathbf{f}_1}[f(\mathbf{x})f(\mathbf{y})] = \prod_{i=1}^{H} \frac{e^{-\frac{[\mu_{*,i}(\mathbf{x}) - \mu_{*,i}(\mathbf{y})]^2}{2(1+\delta_i^2)}}}{\sqrt{1 + \delta_i^2}} \,, \tag{7}$$

*where the conditional means $\mu_{*,i}(\mathbf{x})$ and $\mu_{*,i}(\mathbf{y})$ are associated with the conditional Gaussian density $p(f_{1,i}(\mathbf{x}), f_{1,i}(\mathbf{y})|\mathbf{z}_{1:p}, u_{i,1:p})$, and the positive value $\delta_i^2 = \Sigma_*(\mathbf{x}, \mathbf{x}) + \Sigma_*(\mathbf{y}, \mathbf{y}) - 2\Sigma_*(\mathbf{x}, \mathbf{y})$.*

### 2.3 Random feature expansion

To connect neural networks and above GPs with SE kernel, the following theorem based on the Bochner's theorem is needed. Its proof was provided in Rahimi & Recht (2008).

**Theorem 1.** *The shallow cosine network (Sopena et al., 1999; Gal & Turner, 2015),*

$$f(\mathbf{x}) = \sqrt{\frac{2}{n}} \sum_{i=1}^{n} w_i \cos[\omega_i \cdot (\mathbf{x} - \mathbf{z}_i) + b_i] \,, \tag{8}$$

*is a random parametric function mapping an input $\mathbf{x} \in \mathbb{R}^D$ to $\mathbb{R}$. The collection of independent and normal weight variables, $w_{1:n} \sim \mathcal{N}(0, \sigma^2)$, and bias $b_{1:n} \sim \text{Unif}[0, \pi]$. In above expression, $\mathbf{z}_{1:n} \in \mathbb{R}^D$ are a set of shift vectors, and are referred to as inducing points in GP literature (Gal & Turner, 2015). The random network has zero mean, and the covariance converges to,*

$$\mathbb{E}[f(\mathbf{x})f(\mathbf{y})] \to \sigma^2 \exp[-\frac{1}{2}(\mathbf{x} - \mathbf{y})^T \Lambda (\mathbf{x} - \mathbf{y})] \,, \tag{9}$$

*in the limit $n \to \infty$ if the random vectors $\{\omega_{1:n} \in \mathbb{R}^D\}$ are samples from a multivariate normal distribution $\mathcal{N}(0, \Lambda)$.*

## 3 Shallow trigonometric network

An alternative for the shallow networks in Eq. (8) which yields the same SE covariance was proposed in (Cutajar et al., 2017). With the feature vectors $\omega_{1:n} \in \mathbb{R}^D$, and the random variables $w_{1:M}^c$ and $w_{1:M}^s$ associated with the cosine and sine postactivation, respectively, we can write the random function as,

$$f(\mathbf{x}) = \frac{1}{\sqrt{n}} \sum_{i=1}^{n} w_i^c \cos(\omega_i \cdot \mathbf{x}) + w_i^s \sin(\omega_i \cdot \mathbf{x}) \,, \tag{10}$$

$$= \mathbf{w} \Phi(\Omega \mathbf{x}) \,, \tag{11}$$

in which the compact notation in the second line has $\mathbf{w} = [w_1^c, w_2^c, \cdots, w_n^c, w_1^s, \cdots, w_n^s] \in \mathbb{R}^{1 \times 2n}$ and $\Omega = [\omega_1, \cdots, \omega_n]^T \in \mathbb{R}^{n \times D}$. Activation here is a doublet which reads $\Phi(\ ) = \begin{pmatrix} \cos(\ ) \\ \sin(\ ) \end{pmatrix}$.

Based on the same argument in Rahimi & Recht (2008), Eq. (10) represents a finite-basis model for random smooth functions whose covariance converges to some fixed form in the limit of large $n$. If the features in $\Omega$ are sampled from a distribution and remain fixed, then one can infer the weight parameters $\mathbf{w}$ given the data (or hyperdata in Lu & Shafto (2021b)) $\mathbf{Z}, \mathbf{u}$, the prior distribution $p(\mathbf{w}) = \mathcal{N}(0, \sigma^2 I_{2n})$, and observation noise variance $\sigma_s^2$. The notation means $\mathbf{Z} = (\, \mathbf{z}_1, \cdots, \mathbf{z}_M \,) \in \mathbb{R}^{D \times M}$ and $\mathbf{u} = (\, u_1, \cdots, u_M \,)^T \in \mathbb{R}^{M \times 1}$. Following the linear Bayesian learning (Rasmussen & Williams, 2006), the posterior reads

$$p(\mathbf{w}|\mathbf{Z}, \mathbf{u}) = \mathcal{N}(\mathbf{w}|\bar{\mathbf{w}}, A^{-1}) \,, \tag{12}$$

with the conditional mean and precision matrix,

$$\bar{\mathbf{w}}^T = \sigma_s^{-2} A^{-1} \Phi(\Omega \mathbf{Z}) \mathbf{u} \,, \tag{13}$$

$$A = \sigma_s^{-2} \Phi(\Omega \mathbf{Z}) \Phi^T(\Omega \mathbf{Z}) + \sigma^{-2} I_{2n} \,, \tag{14}$$

where the postactivation matrix $\Phi(\Omega \mathbf{Z})$ has shape $(2n, M)$. Furthermore, the distribution over the predicted value at a new input, $y_* = \mathbf{w}\Phi(\Omega \mathbf{x}_*)$, is still a Gaussian with mean,

$$\bar{f}_* = K_* (\sigma_s^2 I_{2n} + K)^{-1} \mathbf{u} \,,$$

and variance

$$\sigma_*^2 = \sigma_s^2 + K_{**} - K_* (\sigma_s^2 I_{2n} + K)^{-1} K_*^T \,,$$

where we have used the kernel expression $K_* = \sigma^2 \Phi_*^T \Phi$, $K_{**} = \sigma^2 \Phi_*^T \Phi_*$ and $K = \sigma^2 \Phi^T \Phi$ (Rasmussen & Williams, 2006). The shorthand notation has $\Phi_* = \Phi(\Omega \mathbf{x}_*)$ and $\Phi = \Phi(\Omega \mathbf{Z})$. The above result is thus an approximation for GP regression.

In the framework of GP regression, one way to enhance the expressive power of the nonparametric model is, ironically, to form a linear combination of different kernels and treat the coefficients as hyper-parameters optimizing the evidence. The classic regression on Mauna Loa dataset in Rasmussen & Williams (2006) adopts the SE kernel along with rational quadratic and periodic kernels. One may also view the spectra mixture kernel (Wilson & Adams, 2013) as a special kernel composition. For Bayesian neural network, on the other hand, the prior function distribution induced by prior parameter distribution (Wilson & Izmailov, 2020; Zavatone-Veth & Pehlevan, 2021) encodes the expressive power. In practice, design of activation in a network was shown to yield good extrapolation (Pearce et al., 2020). In the following two subsections, we shall introduce two ideas improving the expressivity associated with the trig network representation of GP.

### 3.1 Features from mixture of Gaussians and spectra mixture kernel

Following the work of (Rahimi & Recht, 2008), one can obtain a shallow trig network representation of GP regression model with SE kernel if the features $\Omega$ are sampled from a normal distribution. Similarly, the GP regression models with Laplacian and Cauchy kernels can have their network representation if the features are sampled from different single-mode distributions. The following lemma show that the model with spectra mixture kernel is equivalent to the shallow trig network if the features are sampled from a mixture of Gaussians.

**Lemma 2.** *If the features are sampled from a mixture of multivariate Gaussians, $\omega_{1:n} \sim \sum_a \pi_a \mathcal{N}(\mu_a, \Lambda_a)$ with positive $\pi$'s, and the weight $\mathbf{w} \sim \mathcal{N}(0, \sigma^2 I_{2n})$, then the covariance of outputs in Eq. (10) converges to the spectrum mixture kernel,*

$$k(\mathbf{x}, \mathbf{y}) = \sigma^2 \sum_a \pi_a \cos[\mu_a^T (\mathbf{x} - \mathbf{y})] e^{-\frac{(\mathbf{x}-\mathbf{y})^T \Lambda_a (\mathbf{x}-\mathbf{y})}{2}} \,, \tag{15}$$

*in the wide network limit $n \to \infty$.*

*Proof.* As the weight parameters are independent, one can easily see that the covariance in the large $n$ limit converges to

$$
\mathbb{E}[f(\mathbf{x})f(\mathbf{y})] \to \sigma^2 \int d\omega \ p(\omega) \cos[\omega \cdot (\mathbf{x} - \mathbf{y})]
$$

$$
= \sigma^2 \mathrm{Re} \sum_a \pi_a \int d\omega \ \mathcal{N}(\omega|\mu_a, \Lambda_a)e^{i\omega\cdot(\mathbf{x}-\mathbf{y})}
$$

$$
= \sigma^2 \sum_a \pi_a \cos[\mu_a \cdot (\mathbf{x} - \mathbf{y})] \exp[-\frac{1}{2}(\mathbf{x} - \mathbf{y})^T \Lambda_a (\mathbf{x} - \mathbf{y})] \, .
$$

In the first equality, Re refers to as the operation of taking real part. $\qquad\square$

### 3.2 Prior distribution over the network output

Here, we investigate the marginal prior function distribution $p(f) = \int d\mathbf{w} \ p(f|\mathbf{w})p(\mathbf{w})$ induced by the prior weight distribution $p(\mathbf{w})$. Following the technique in Remark 1, we can conclude that the prior function distribution associated with the shallow trig network in Eq. (10) is Gaussian, independent of the feature number $n$.

**Remark 2.** *The probability density over the function Eq. (10) for a single input is always a Gaussian with zero mean and variance $\sigma^2$, independent of the width $n$ and of the sampling distribution $p(\Omega)$.*

*Proof.* Given $\mathbf{w}$ is independent normal with variance $\sigma^2$, the conditional distribution $p(f|\Phi)$ is also a normal with variance $\frac{\sigma^2}{n}\sum_{i=1}^n \cos^2 \omega_1 \cdot \mathbf{x} + \cdots \cos^2 \omega_n \cdot \mathbf{x} + \sin^2 \omega_1 \cdot \mathbf{x} + \cdots + \sin^2 \omega_n \cdot \mathbf{x} = \sigma^2$. Thus, $p(f(\mathbf{x})) = \mathcal{N}(0, \sigma^2)$. $\qquad\square$

It was suggested that the superior expressive power of deep linear network and ReLu network is related to their non-Gaussian prior function distribution (Vladimirova et al., 2019; Roberts et al., 2021; Zavatone-Veth & Pehlevan, 2021). Besides the network with finite width which lifts the Gaussianity (Yaida, 2020), the following shallow network $f_\psi : \mathbb{R}^D \mapsto \mathbb{R}$ with modified preactivation,

$$
f_\psi(\mathbf{x}) = \frac{1}{\sqrt{n}} \sum_{i=1}^n w_i^c \cos[\omega_i \cdot \mathbf{x} + \psi(\mathbf{x})] + w_i^s \sin[\omega_i \cdot \mathbf{x} - \psi(\mathbf{x})] \, , \tag{16}
$$

incorporating a phase shift network $\psi(\mathbf{x})$ is proposed to lift the Gaussianity.

**Lemma 3.** *The Fourier transformed $\tilde{p}(q)$ associated with the prior distribution over the output in Eq. (16) is,*

$$
\tilde{p}(q) = e^{-\frac{1}{2}q^2\sigma^2} \int d\omega p(\omega) e^{\frac{1}{2}q^2\sigma^2 \sin\psi(\mathbf{x})\sin 2\omega\cdot\mathbf{x}} \, , \tag{17}
$$

*where the feature $\omega \in \mathbb{R}^D$ are sampled from the high dimensional normal distribution $p(\omega) = \prod_{d=1}^D \mathcal{N}(\omega_d|0, \sigma_d^2)$.*

It can be seen that the phase shift network $\psi(\mathbf{x})$ lifts the Gaussian character of the prior distribution, but the intractable high-dimensional integral in Eq. (17) stands in the way of obtaining a closed form for its characteristic function. Nevertheless, one can proceed with the approximation of Gauss-Hermite quardature (Greenwood & Miller, 1948). Consider the case where the variances $\sigma_{1:D}^2 = \sigma_F^2$ associated with the features in all dimensions are the same, and after including the most relevant terms,

$$
\tilde{p}(q) \approx e^{-\frac{1}{2}q^2\sigma^2}(\frac{\lambda_0}{\sqrt{\pi}})^D \{1 + 2\frac{\lambda_1}{\lambda_0}\sum_{d=1}^D \cosh[\frac{1}{2}q^2\sigma^2 \sin\psi(\mathbf{x})\sin(2\sqrt{2}\sigma_F z_1 x_d)]\} \, , \tag{18}
$$

where the coefficients $\lambda_0 \approx 1.181$ and $\lambda_1 \approx 0.295$ are given in (Greenwood & Miller, 1948) and $z_1 \approx 1.22$ is the nonzero root of the third order Hermite polynomial. Consequently, the characteristic $\tilde{p}$ obtains a non-Gaussian correction $\propto q^4 e^{-q^2\sigma^2}$ for small Fourier component $q$.

## 4 Deep trigonometric network

Now we proceed to consider the deep trigonometric network proposed in Cutajar et al. (2017). With the same notation as the shallow network, the deep trigonometric network of interest has the following matrix representation,

$$f(\mathbf{x}) = \mathbf{w}_2 \Phi(\Omega_2 \mathbf{W}_1 \Phi(\Omega_1 \mathbf{x})) , \tag{19}$$

in which the random weight matrices $\mathbf{w}_2 \in \mathbb{R}^{1 \times 2n_2}$, $\mathbf{W}_1 \in \mathbb{R}^{H \times 2n_1}$ and the feature matrices $\Omega_2 \in \mathbb{R}^{n_2 \times H}$, $\Omega_1 \in \mathbb{R}^{n_1 \times D}$. Here, the hidden output $\mathbf{h} = \mathbf{W}_1 \Phi(\Omega_1 \mathbf{x})$ has bottleneck (Agrawal et al., 2020) dimension $H$ collecting the $n_1$ postactivations. Besides the compositional hierarchy which makes the function more expressive than its shallow counterpart, one can also adopt different prior distribution over the weight and feature matrices. In the following three subsections, we shall discuss the cases of (i) the entries in $\mathbf{W}_1$, $\mathbf{w}_2$, $\Omega_1$, and $\Omega_2$ are all independent normal, which corresponds to the zero-mean two-layer DGP with SE kernels (Lu et al., 2020), (ii) same as in (i) but in the first layer the weight entries in $\mathbf{W}_1 \sim p(\mathbf{w}_1 | \mathbf{Z}, \mathbf{U})$ are not independent, which corresponds to the two-layer DGP for multi-fidelity regression (Lu & Shafto, 2021a) and hyper-data learning (Lu & Shafto, 2021b) with $\mathbf{Z}, \mathbf{U}$ acting as the support in the latent function, and (iii) same as in (ii) but the feature matrix $\Omega_2$ consists of samples from the mixture of Gaussians, which corresponds to the two-layer DGP with outer GP using the SM kernel.

### 4.1 Deep trig net covariance and random matrix spectrum

To show that the deep trigonometric network yields the same covariance as the two-layer DGP when the entries in weight matrices have independent normal distribution, the spectrum of the following square random matrix with dimension $2n_1$,

$$G = [\Phi(\Omega_1 \mathbf{x}) - \Phi(\Omega_1 \mathbf{y})][\Phi(\Omega_1 \mathbf{x}) - \Phi(\Omega_1 \mathbf{y})]^T , \tag{20}$$

is critical in determining the statistics of network outputs.

**Remark 3.** *The square matrix $G$ has $(2n_1 - 1)$ zero eigenvalues and one nonzero eigenvalue. If the set of feature vectors $\{\omega_{1:n_1}\}$ are sampled from $\mathcal{N}(\omega | 0, I_D)$, then the nonzero eigenvalue shall converge to the following,*

$$\left| \Phi(\mathbf{x}) - \Phi(\mathbf{y}) \right|^2 = \frac{1}{n_1} \sum_{i=1}^{n_1} (\cos \omega_i \mathbf{x} - \cos \omega_i \mathbf{y})^2 + (\sin \omega_i \mathbf{x} - \sin \omega_i \mathbf{y})^2$$

$$\rightarrow 2 - 2k_{\text{SE}}(\mathbf{x}, \mathbf{y}) ,$$

*in the limit $n_1 \rightarrow \infty$. $k_{\text{SE}}(\mathbf{x}, \mathbf{y}) = \exp[-\frac{1}{2}|\mathbf{x} - \mathbf{y}|^2]$ stands for the squared exponential covariance function with all hyper-parameters set to unity.*

*Proof.* First, one can view $\mathbf{v} = \Phi(\Omega_1 \mathbf{x}) - \Phi(\Omega_1 \mathbf{y})$ as a fixed vector in the $2n_1$ dimensional space. The entries read $\frac{1}{\sqrt{n_1}} [\cos \omega_1 \cdot \mathbf{x} - \cos \omega_1 \cdot \mathbf{y}, \cdots, \cos \omega_{n_1} \cdot \mathbf{x} - \cos \omega_{n_1} \cdot \mathbf{y}, \sin \omega_1 \cdot \mathbf{x} - \sin \omega_1 \cdot \mathbf{y}, \cdots]$. Then, one can in principle find the orthogonal subspace, spaned by the set of vectors $\{\mathbf{v}_{\perp, 1:(2n_1-1)}\}$, to $\mathbf{v}$ in the space. Thus, we have $\mathbf{v}^T \mathbf{v}_\perp = 0$, one can write the zero eigenvalue equations,

$$G\mathbf{v}_\perp = \mathbf{v}\mathbf{v}^T \mathbf{v}_\perp = 0 ,$$

and the only nonzero eigenvalue one,

$$G\mathbf{v} = |\mathbf{v}|^2 \mathbf{v} .$$

$\square$

With the knowledge of the spectrum of random matrix $G$, now we can go on to derive the desired covariance of deep trigonometric network.

**Lemma 4.** *The covariance of the deep trigonometric network in Eq. (19),*

$$\mathbb{E}_{\mathbf{W}_1}\left\{\mathbb{E}_{\mathbf{w}_2|\mathbf{W}_1}\left[f(\mathbf{x})f(\mathbf{y})\right]\right\} \rightarrow k_{\text{DGP}}(\mathbf{x},\mathbf{y}) = \left\{1 + 2[1 - k_{\text{SE}}(\mathbf{x},\mathbf{y})]\right\}^{-\frac{H}{2}}, \tag{21}$$

*as the numbers of features $n_1$ and $n_2$ both approach infinity.*

*Proof.* The independence among the zero-mean random weights $\mathbf{w}_2$ and uniform variance leads to $\mathbb{E}_{\mathbf{w}_2}[(\mathbf{w}_2\Phi(\Omega_2\mathbf{h}_x))(\mathbf{w}_2\Phi(\Omega_2\mathbf{h}_y))] = \Phi(\Omega_2\mathbf{h}_x)^T\Phi(\Omega_2\mathbf{h}_y)$, which at the limit $n_2 \rightarrow \infty$ results in,

$$\mathbb{E}[f(\mathbf{x})f(\mathbf{y})] \rightarrow \mathbb{E}_{\mathbf{W}_1}\left[e^{-\frac{d^2(\mathbf{x},\mathbf{y})}{2}}\right],$$

where the squared distance between the latent outputs $\mathbf{h}(\mathbf{x})$ and $\mathbf{h}(\mathbf{y})$ in the exponent can be rewritten as,

$$\begin{aligned}
d^2(\mathbf{x},\mathbf{y}) &= [\mathbf{h}(\mathbf{x}) - \mathbf{h}(\mathbf{y})]^T[\mathbf{h}(\mathbf{x}) - \mathbf{h}(\mathbf{y})] \\
&= \text{Tr}\left\{\mathbf{W}_1[\Phi(\Omega_1\mathbf{x}) - \Phi(\Omega_1\mathbf{y})][\Phi(\Omega_1\mathbf{x}) - \Phi(\Omega_1\mathbf{y})]^T\mathbf{W}_1^T\right\} \\
&= \sum_{i=1}^{H}\mathbf{w}_{1,i}G\mathbf{w}_{1,i}^t,
\end{aligned}$$

where the rows of $\mathbf{W}_1$ are written as $\{\mathbf{w}_{1,1:H}\}$ in the last line. Lastly, the determinant of $(I_{2n_1} + G)$ enters as a result of

$$\mathbb{E}_{\mathbf{w}_{1,1:H}\sim\mathcal{N}(0,I_{2n_1})}[e^{-d^2(\mathbf{x},\mathbf{y})}] = \Pi_{i=1}^{H}\frac{1}{\sqrt{\det[I_{2n_1} + G]}}. \tag{22}$$

$\square$

## 4.2 Deep trig net with weights representing latent function support

In above subsection, the deep trigonometric net with centered and independent Gaussian weights $\mathbf{W}_1$ and $\mathbf{w}_2$ is equivalent to composition of two zero-mean GPs. In Lu & Shafto (2021b), it was shown that treating the support in latent function, i.e. $M$ hyper-data points with $\mathbf{h}(\mathbf{z}_{1:M}) = \mathbf{u}_{1:M}$, as additional hyper-parameters can enhance generalization of DGPs. $\mathbf{z} \in \mathbb{R}^D$ and $\mathbf{u} \in \mathbb{R}^H$. Here, the function space view translates to the weight parameters, $\mathbf{W}_1|\mathbf{Z},\mathbf{U} \sim \prod_{i=1}^{H}\mathcal{N}(\mathbf{w}_{1,i}|\bar{\mathbf{w}}_i, A^{-1})$, conditional on the hyper-input and output matrices, $\mathbf{Z} := (\mathbf{z}_1, \cdots, \mathbf{z}_M) \in \mathbb{R}^{D\times M}$ and $\mathbf{U} = (\mathbf{u}_1, \cdots, \mathbf{u}_M) \in \mathbb{R}^{H\times M}$, respectively. The conditional precision matrix,

$$A = \left[I_{2n_1} + \Phi(\Omega_1\mathbf{Z})\Phi^T(\Omega_1\mathbf{Z})\right] \tag{23}$$

and the conditional mean for each output dimension,

$$\bar{\mathbf{w}}_{1,i} = A^{-1}\Phi(\Omega_1\mathbf{Z})\mathbf{U}_{i,:}^T, \tag{24}$$

which can be found in Ch.2.1.2 in Rasmussen & Williams (2006) [also in Ober & Aitchison (2021)].

**Lemma 5.** *If the latent layer weights $\mathbf{W}_1$ in Eq. (19) have the correlated prior distribution $\mathbf{W}_1 \sim \prod_{i=1}^{H}p(\mathbf{w}_{1,i}|\mathbf{Z},\mathbf{U}_{i,:})$, then the covariance converges to the DGP covariance in Eq. (7).*

*Proof.* The proof follows the previous one except that we are evaluating the following expectation,

$$\begin{aligned}
\mathbb{E}[f(\mathbf{x})f(\mathbf{y})] &= \mathbb{E}_{\mathbf{w}_{1,1:H}\sim\mathcal{N}(\bar{\mathbf{w}}_{1:H},A^{-1})}\left[e^{-\frac{\mathbf{w}_{1,1}G\mathbf{w}_{1,1}^T}{2}}e^{-\frac{\mathbf{w}_{1,2}G\mathbf{w}_{1,2}^T}{2}}\cdots e^{-\frac{\mathbf{w}_{1,H}G\mathbf{w}_{1,H}^T}{2}}\right] \\
&= \prod_{i=1}^{H}\frac{e^{-\frac{1}{2}\bar{\mathbf{w}}_i(I+GA^{-1})^{-1}G\bar{\mathbf{w}}_i^T}}{\sqrt{|I + A^{-1}G|}},
\end{aligned}$$

in which we just focus on one term in the product. By writing the matrix $G = \mathbf{v}\mathbf{v}^T$ related to the inputs $\mathbf{x},\mathbf{y}$ (see Remark 7) and using the matrix inversion lemma, the exponent in above expression becomes

$-\frac{1}{2}\bar{\mathbf{w}}_i\mathbf{v}(1+\mathbf{v}^TA^{-1}\mathbf{v})^{-1}\mathbf{v}^T\bar{\mathbf{w}}_i^T$. As for the determinant in denominator, the matrix $A^{-1}G$ does not couple the vector $\mathbf{v}$ with its orthogonal subspace $\mathbf{v}_\perp$, leading to $|I+A^{-1}G|=1+(\mathbf{v}^TA^{-1}G\mathbf{v})/(\mathbf{v}^T\mathbf{v})$. With some manipulation and lengthy calculation,

$$\left|I+A^{-1}G\right| = 1 + [\Phi_x - \Phi_y]^T[\Phi_x - \Phi_y] - [\Phi_x - \Phi_y]^T\Phi_Z[I+\Phi_Z^T\Phi_Z]^{-1}\Phi_Z^T[\Phi_x - \Phi_y]$$
$$\to 1 + k_{xx} + k_{yy} - 2k_{xy} - k_{xZ}K_{ZZ}^{-1}k_{Zx} - k_{yZ}K_{ZZ}^{-1}k_{Zy} + 2k_{xZ}K_{ZZ}^{-1}k_{Zy}\,.$$

It can also be seen that the above result is identical to $(1+\mathbf{v}^TA^{-1}\mathbf{v})$. Similarly, one can show the scalar $\bar{\mathbf{w}}_i\mathbf{v}\mathbf{v}^T\bar{\mathbf{w}}_i^T = (m_x - m_y)^2$ with the limiting form $m_x \to k_{xZ}K_{ZZ}^{-1}\mathbf{U}_{i,:}$. $\qquad\square$

### 4.3 Deep trig net with mixed spectrum features

The deep trigonometric networks are expressive as the choices over the weights' prior distribution are flexible. One may also consider employing different distributions over the features as we do in the shallow nets. Here, we are interested in the resultant covariance when the outer features $\Omega_2$ consist of samples from mixture of Gaussians at different centers and the inner weights $\mathbf{W}_1$ representing the latent function support.

**Lemma 6.** *When the deep trigonometric network in Eq. (19) has fixed features $\omega_2 \in \mathbb{R}$ from samples of a mixed distribution $\sum_a \pi_a \mathcal{N}(\mu_a, \sigma_a^2)$ and the random variables $\mathbf{w}_1$ represent the weight space view of latent function support $\mathbf{w}_1\Phi(\Omega_1\mathbf{z}_{1:M}) = u_{1:M}$, it is equivalent to the DGP $f(\mathbf{x}) = f_2(f_1(\mathbf{x}))$ with $f_1|\mathbf{Z},\mathbf{u} \sim \mathcal{GP}(m,\Sigma)$ and $f_2|f_1 \sim \mathcal{GP}(0, k_{SM})$. $m$ and $\Sigma$ are the conditional mean and covariance matrix given the hyper-data $\mathbf{Z},\mathbf{u}$. The covariance is,*

$$\mathbb{E}[f(\mathbf{x})f(\mathbf{y})] = \sum_a \frac{\pi_a}{(1+\sigma_a^2\delta^2)^{1/2}} \exp\Big[-\frac{\sigma_a^2(m_x-m_y)^2 + \delta^2\mu_a^2}{2(1+\sigma_a^2\delta^2)}\Big] \cos\Big[\frac{\mu_a(m_x-m_y)}{1+\sigma_a^2\delta^2}\Big]\,. \qquad (25)$$

*Proof.* It is easier to work out the covariance in the function space. Observing that

$$\mathbb{E}_{f_1|\mathbf{Z},\mathbf{u}}\big\{\mathbb{E}_{f_2|f_1}[f_2(f_1(\mathbf{x}))f_2(f_1(\mathbf{y}))]\big\} = \operatorname{Re}\mathbb{E}_{f_1|\mathbf{Z},\mathbf{u}}[\mathbb{E}_{\omega_2}\,e^{i\omega_2[f_1(\mathbf{x})-f_1(\mathbf{y})]}]\,,$$

one can compute the expectation with respect to the latent function $f_1$ first, followed by that of feature $\omega_2$. Thus, we get the covariance,

$$\mathbb{E}_{\omega_2\sim\sum_a\pi_a\mathcal{N}(\mu_a,\sigma_a^2)}\big\{\mathbb{E}_{(f_1(\mathbf{x}),f_1(\mathbf{y}))^T\sim\mathcal{N}(m,\Sigma)}[\cos\omega_2(f_1(\mathbf{x})-f_1(\mathbf{y}))]\big\}\,,$$

which can be computed analytically. $\qquad\square$

Such deep trigonometric network is closely related to the deep kernel learning with the SM kernel (Wilson et al., 2016). Now it becomes clear that the outer network represents the random function $f_2 \sim \mathcal{GP}(0, k_{SM})$. The hyper-data $\mathbf{Z},\mathbf{u}$ constrain the inner function $f_1$, and in the limit when the hyper-data are dense the function $\mathbf{f}_1$ becomes deterministic (Lu & Shafto, 2021b). Such situation is equivalent to passing the inputs to a parametric function and then to a GP. However, the probabilistic nature in $f_1$ in the sparse hyper-data limit is helpful for preventing overfitting in deep kernel learning with over-parameterized $f_1$ (Ober et al., 2021).

## 5 Neural tangent kernel for trigonometric networks

For probabilistic regression problems with data matrix $\mathbf{X}$ and observations $\mathbf{y}$, one has two choices over the models for prediction. The first choice is function-based models, such as GPs and DGPs. The exact GP inference produces a predictive distribution $p(y_*|\mathbf{X},\mathbf{y},\mathbf{x}_*)$ with closed form mean and variance that only depends on the covariance function and hyper-parameters. However, such luxury is not carried over to DGP as there is no corresponding exact inference. The second choice is weight-based models: the shallow Bayesian neural network, Eq. (10), and its deep version, Eq. (19). For shallow trig network with fixed feature matrix $\Omega$, then it becomes a Bayesian linear regression problem (see Sec. 3), and the predictive mean and variance converge to the GP's result as the number of features $n \to \infty$.

It is not clear whether the appealing correspondence between shallow trig network and GP with SE kernel can carry to the deep trigonometric network and 2-layer DGP discussed in this paper. Nevertheless, the perspective from neural tangent kernel (Jacot et al., 2018; Arora et al., 2019) may shed some light on this issue. For gradient based learning of infinite and deep neural network $f(\mathbf{x}|\theta)$, the network function shall eventually converge to the predictive mean of GP with the following kernel,

$$k(\mathbf{x}, \mathbf{y}) = \mathbb{E}_\theta \Big[ \frac{\partial f(\mathbf{x}|\theta)}{\partial \theta} \cdot \frac{\partial f(\mathbf{y}|\theta)}{\partial \theta} \Big] \,, \tag{26}$$

where the derivative operation $\partial/\partial\theta$ with respect to all weight parameters in $\theta$ generates a vector. Moreover, the neural tangent kernel remains a constant during the gradient descent, so its value is determined by the initial distribution over $\theta$ (a recent study (Seleznova & Kutyniok, 2021) suggested that the neural tangent kernels for deeper model may still evolve during training).

Now, given the fact that the deep trigonometric network yields the same covariance as the two-layer DGP with SE kernels, it is interesting to derive the neural tangent kernel associated with Eq. (19), which may reveal some insights into the correspondence between deep trigonometric network and DGP.

**Lemma 7.** *Assume that the features $\Omega_{1,2}$ in the deep trigonometric network in Eq. (19) are fixed and the weights $\mathbf{w}_{1,2}$ are learned through gradient descent with squared loss function. Then the associated neural tangent kernel reads,*

$$k_{\mathrm{NTK}}(\mathbf{x}, \mathbf{y}) = k_{\mathrm{DGP}} + k_{\mathrm{SE}} k_{\mathrm{DGP}}^3 \,, \tag{27}$$

*where $k_{\mathrm{SE}}$ is the SE covariance function and $k_{\mathrm{DGP}}$ is the exact covariance of the two-layer DGP. Note that we have set all hyper-parameters to unit for ease of notation.*

*Proof.* As only the weight parameters are learned, the neural tangent kernel has the following expression,

$$k_{NTK}(\mathbf{x}, \mathbf{y}) = \mathbb{E}_{\mathbf{W}_1} \left\{ \mathbb{E}_{\mathbf{W}_2|\mathbf{W}_1} \Big[ \frac{\partial f(\mathbf{x})}{\partial \mathbf{W}_2} \frac{\partial f(\mathbf{y})}{\partial \mathbf{W}_2} + \frac{\partial f(\mathbf{x})}{\partial \mathbf{W}_1} \frac{\partial f(\mathbf{y})}{\partial \mathbf{W}_1} \Big] \right\}$$
$$= K_{\mathrm{DGP}} + K_e \,,$$

where we observe that the first term (derivative wrt second weight $\mathbf{w}_2$) is the same as the covariance of DGP (see Sec. 4.1). Next, we shall focus on the second term, $K_e$. Notice that the order of differentiation $\partial f/\mathbf{w}_1$ and the expectation $\mathbb{E}_{\mathbf{w}_2|\mathbf{w}_1}$ can be switched. To facilitate the computation, we can temporarily write $f(\mathbf{x}) = \mathbf{w}_2 \Phi(\Omega_2 \mathbf{w}_a \Phi(\Omega_1 \mathbf{x}))$ and $f(\mathbf{y}) = \mathbf{w}_2 \Phi(\Omega_2 \mathbf{w}_b \Phi(\Omega_1 \mathbf{y}))$ so that we can first compute the expectation and then take the derivatives. The rest of derivations just rest on some simple tricks,

$$K_e = \mathbb{E}_{\mathbf{w}_1} \left\{ \sum_{i=1}^{n_1} \frac{\partial^2}{\partial w_{a,i} \partial w_{b,i}} \mathbb{E}_{\mathbf{w}_2|\mathbf{w}_{a,b}} [f_a(\mathbf{x}) f_b(\mathbf{y})]\big|_{\mathbf{w}_a = \mathbf{w}_b = \mathbf{w}_1} \right\}$$
$$= \mathbb{E}_{\mathbf{w}_1} \left[ e^{-\frac{\mathbf{w}_1 G \mathbf{w}_1^T}{2}} \Phi^T(\Omega_1 \mathbf{x}) \Phi(\Omega_1 \mathbf{y}) (1 - \mathbf{w} G \mathbf{w}_1^T) \right]$$
$$= e^{-\frac{|\mathbf{x}-\mathbf{y}|^2}{2}} (1 + 2\frac{\partial}{\partial \lambda}) \mathbb{E}_{\mathbf{w}_1} \left[ e^{-\lambda \frac{\mathbf{w}_1 G \mathbf{w}_1^T}{2}} \right]\big|_{\lambda=1}$$
$$= k_{\mathrm{SE}}(\mathbf{x}, \mathbf{y}) \big[ 1 + 2(1 - k_{\mathrm{SE}}(\mathbf{x}, \mathbf{y})) \big]^{-\frac{3}{2}} \,.$$

To arrive at the second equality, we have used

$$\sum_i \partial^2_{w_{a,i} w_{b,i}} e^{-\frac{1}{2}(\mathbf{w}_a \cdot \Phi_x - \mathbf{w}_b \cdot \Phi_y)^2} = e^{-\frac{1}{2}(\mathbf{w}_a \cdot \Phi_x - \mathbf{w}_b \cdot \Phi_y)^2} (\Phi_x \cdot \Phi_y) \big[ 1 - (\mathbf{w}_a \cdot \Phi_x - \mathbf{w}_b \cdot \Phi_y)^2 \big] \,.$$

$\square$

As for the shallow trigonometric net in Eq. (10), it is easy to show that the NTK is the same as $k_{\mathrm{SE}}$ if $\Omega$ are independent and normal. Hence, the predictive distribution for $y_*|\mathbf{x}_*, \Omega, \mathbf{X}, \mathbf{y}$ is the same for the shallow Bayesian trigonometric network in wide limit and GP with SE kernel. Moreover, the mean of this distribution

shall coincide with the prediction obtained using gradient descent as the equivalence between NTK and $k_{\text{SE}}$ suggests. However, the correspondence between DGP and deep trigonometric network is intriguing as there is no exact inference for both models. If one adopts the moment matching inference (Lu et al., 2020) which treats the marginal prior distribution of DGP as a GP (Lu & Shafto, 2021b), then the predictive distribution is the same as the GP with $k_{\text{DGP}}$. With the equivalence between DGP and deep trigonometric net, one can say that the single prediction made by gradient descent algorithm shall converge to the predictive mean of a GP with $k_{\text{NTK}}$ in Eq. (27). The origin for the discrepancy between $k_{\text{DGP}}$ and $k_{\text{NTK}}$ is a very interesting question as the exact DGP inference is impossible and the optimization of deep trigonometric network is not convex.

## 6 Finite width corrections

For both the shallow and deep is much intriguing networks, their output $f(\mathbf{x})$ depend on two sets of parameters: the weights $\mathbf{W}$'s and the projections $\Omega$'s. We have connected them with shallow GPs and deep GPs, respectively. By treating the layer widths to be infinity, we have obtained the limiting kernel $k_{\text{DGP}}$ and neural tangent kernel $k_{\text{NTK}}$ for the deep network. Here, we shall consider the deviation from the limiting kernels when the layer width is large but finite. An important implication is that the kernel only converges to its fixed and limiting form when the inner width $n_1$ is infinite, suggesting that the inner layer is more relevant to the feature learning than the outer one.

We follow (Yu et al., 2016) and define the kernel estimator, $\hat{k}_{\text{DGP}}(\mathbf{x}, \mathbf{y}) := \mathbb{E}_{\mathbf{w}}[f(\mathbf{x})f(\mathbf{y})|\Omega]$, for the deep net. With some simple algebra,

$$\hat{k}_{\text{DGP}} = \frac{1}{n_2}\text{Re}\sum_i \mathbb{E}_{\mathbf{W}^{(1)}}\{\prod_{k,m}\exp[i\omega_{ik}^{(2)}w_{km}^{(1)}(\Phi_{\mathbf{x}} - \Phi_{\mathbf{y}})_m]\} \tag{28}$$

$$= \frac{1}{n_2}\sum_{i=1}^{n_2}\exp\{-\sigma_w^2\sum_{k=1}^{H}[\omega_{ik}^{(2)}]^2 \cdot \frac{1}{n_1}\sum_{m=1}^{n_1}\left[1 - \cos\Omega_m^{(1)} \cdot (\mathbf{x} - \mathbf{y})\right]\} \tag{29}$$

where the components of post-activation vector read $\Phi_{\mathbf{x}} = (1/\sqrt{n_1})[\cos\Omega_{1:n_1}^{(1)} \cdot \mathbf{x}, \sin\Omega_{1:n_1}^{(1)} \cdot \mathbf{x}]$, and the above second equality follows from the fact that weights $w^{(1)} \sim \mathcal{N}(0, \sigma_w^2)$ are iid. The two summations are over the projection parameters $\omega^{(2)}$ in outer layer and projection vectors $\Omega^{(1)}$ in inner layer. Now the estimator $\hat{k}_{\text{DGP}}$ depends on the realizations of $\Omega_{1:n_1}^{(1)}$ and $\Omega_{1:n_2}^{(2)}$.

**Lemma 8.** *When the latent dimension $H$ is finite and the inner layer width $n_1$ is large but finite, the mean of kernel estimator for the deep trig network approximately reads,*

$$\mathbb{E}_{\Omega}[\hat{k}_{\text{DGP}}(\mathbf{x}, \mathbf{y})] \approx [\int e^{-(1-\hat{k}_{\text{SE}})\omega^2\sigma_w^2}d\mu(\omega)d\mu(\hat{k}_{\text{SE}})]^H \tag{30}$$

*with the normal $\omega$ representing iid entries in $\Omega_{1:n_2}^{(2)}$ and $\hat{k}_{\text{SE}} := (1/n_1)\sum_m \cos\Omega_m^{(1)} \cdot (\mathbf{x} - \mathbf{y})$. Here $d\mu(\hat{k}_{\text{SE}})$ takes the approximate density $\mathcal{N}(\mu_s, \sigma_s^2)$ with mean $\mu_s := \mathbb{E}_{\Omega^{(1)}}[\hat{k}_{\text{SE}}]$ and variance $\sigma_s^2 := \text{Var}_{\Omega^{(1)}}[\hat{k}_{\text{SE}}]$.*

*Proof.* First, rewriting the expectation of some smooth function $\alpha$ as $\mathbb{E}_{\Omega^{(1)}}[\alpha(\hat{k}_{\text{SE}})] = \mathbb{E}_{\hat{k}_{\text{SE}}}[\alpha(\hat{k}_{\text{SE}})]$ is valid so one can apply it to Eq. (29) as well. Next, $\hat{k}_{\text{SE}}$ has mean $\mu_s = k_{\text{SE}}$ and variance $\sigma_s^2 = (1 - k_{\text{SE}}^2)^2/(2n_1)$ if $\Omega^{(1)}$ is normal (Yu et al., 2016). For large but finite $n_1$, the central limit theorem suggests that $\hat{k}_{\text{SE}}$ can be treated as a Gaussian. Lastly, the iid and normal assumption of entries in $\Omega^{(2)}$ result in the product form. $\qquad\square$

A few observations follow from the lemma. First, when $n_1$ is infinite, the random variable $\hat{k}_{\text{SE}}$ becomes deterministic as $\sigma_s^2$ vanishes [(Lee et al., 2018) employed similar strategy in proving GP behavior for DNNs]. Thus the density $d\mu(\hat{k}_{\text{SE}})$ approaches a delta function and the remaining integration over $\omega$ results in $\mathbb{E}[\hat{k}_{\text{DGP}}] = k_{\text{DGP}}$. Note that, due to the randomness in $\omega$, $\text{Var}[\hat{k}_{\text{DGP}}]$ does not vanish, which signifies the difference with NNGP. Secondly, when the latent dimension $H$ is also infinite and when the weight

variance has $\sigma_w^2 = 1/H$, then the term $\sigma_w^2 \sum_{k=1}^{H} [\omega_{ik}^{(2)}]^2$ summing over squared projection parameters in outer layer in Eq. (29) also converges to its fixed mean, which in turn leads to $\mathbb{E}[\hat{k}_{\mathrm{DGP}}] = \exp[k_{\mathrm{SE}} - 1]$ along with vanishing $\mathrm{Var}[\hat{k}_{\mathrm{DGP}}]$. This limiting kernel first appeared in (Duvenaud et al., 2014) discussing asymptotic kernel of DNNs, while it corresponds to the case when the variances in $\hat{k}_{\mathrm{SE}}$ and $\overline{\omega^2}$ both vanish.

As for finite $n_{1,2}$ and $H$, one can proceed to show $\mathbb{E}[\hat{k}_{\mathrm{DGP}}] = \langle [1 + 2\sigma_w^2(1 - \hat{k}_{\mathrm{SE}})]^{-1/2} \rangle^H$ after marginalizing the entries in $\Omega_{1:n_2}^{(2)}$. The brackets $\langle \cdot \rangle$ stands for averaging wrt the random variable $\hat{k}_{\mathrm{SE}}$. However, even with $\hat{k}_{\mathrm{SE}}$ approximately being a Gaussian, the mean does not have a closed form. Nevertheless, we again employ the Gauss-Hermite quardature method to approximate the integration. The following remark summarizes the deviation from the limiting $k_{\mathrm{DGP}}$ due to the finite width $n_{1,2}$.

**Remark 4.** *Consider $H = 1$, one can show the approximate deviation yields,*

$$|k_{\mathrm{DGP}} - \mathbb{E}_{\Omega}[\hat{k}_{\mathrm{DGP}}]| \approx \frac{3\lambda_1 z_1^2 \sigma_w^4}{n_1 \sqrt{\pi}} (1 - k_{\mathrm{SE}}^2)^2 k_{\mathrm{DGP}}^3 , \tag{31}$$

*in which the values of Gauss-Hermite parameters $\lambda_{0,1}$ and $z_1$ are listed in (Greenwood & Miller, 1948).*

*Proof.* Considering contributions from the three roots $\{z_0, z_\pm\}$ of the third order Hermite polynomial, the approximation of integral reads $\mathbb{E}[\hat{k}_{\mathrm{DGP}}] = \sum_{i=0,\pm 1} \frac{\lambda_i}{\sqrt{\pi}} [1 + 2\sigma_w^2(1 - k_{\mathrm{SE}} + \sqrt{2}\sigma_s z_i)]^{-1/2}$, which, for zeroth order of $\sigma_s$, gives $\mathbb{E}[\hat{k}_{\mathrm{DGP}}] = (\lambda_0 + 2\lambda_1)k_{\mathrm{DGP}}/\sqrt{\pi}$ where the fact $\lambda_1 = \lambda_{-1}$ is used. The next order of correction is $O(\sigma_s^2)$ due to the symmetry $z_1 = -z_{-1}$ and the expansion $(1 + \epsilon)^{-1/2} = 1 - \epsilon/2 + 3\epsilon^2/8 + \cdots$. One can thus recover the above expression if one further takes $(\lambda_0 + 2\lambda_1)/\sqrt{\pi} \approx 0.99918$ to be unity. $\square$

It is interesting to note from the minimum deep model the nontrivial effect of depth on statistics of $\hat{k}_{\mathrm{DGP}}$. For the shallow model in (Yu et al., 2016), the mean coincides with the *fixed* kernel, i.e. $\mathbb{E}[\hat{k}_{\mathrm{SE}}] = k_{\mathrm{SE}}$. In contrast, $\mathbb{E}[\hat{k}_{\mathrm{DGP}}] \neq k_{\mathrm{DGP}}$ when the inner width $n_1$ is not infinite, which implies that the inner layer is more relevant to feature learning than the outer one. Aitchison (2020) had similar observation in a two-layer linear Bayesian model.

The same formulation can be applied to analytically investigate the finite-width effect on NTK. After some manageable algebra, we can arrive the following estimator for NTK,

$$\hat{k}_{\mathrm{NTK}} = \frac{1}{n_2} \sum_{i=1}^{n_2} \{1 + [\omega_i^{(2)}]^2 + \frac{\partial}{\sigma_w^2 \partial \lambda} \} e^{-\lambda \sigma_w^2 [\omega_i^{(2)}]^2 (1 - \hat{k}_{\mathrm{SE}})} \big|_{\lambda=1} , \tag{32}$$

for $H = 1$. The deviation $|\hat{k}_{\mathrm{NTK}} - k_{\mathrm{NTK}}| \approx (6\lambda_1/\sqrt{\pi})(\sigma_s^2 \sigma_w^2 z_1^2 k_{\mathrm{DGP}}^2)(1 + 2\sigma_w^2) \propto (1/n_1)$ can be obtained by similar computations. The NTK case of deep ReLu network was studied in (Hanin & Nica, 2019) but with a rather different approach and assumption.

# 7 Simulations

In this paper, an important consequence of the translation between DGP in weight representation and function representation is that one can pursue the MAP estimate of weight parameters from the exact posterior. The point estimate then allows to obtain the mean of predictive prediction, which does not seem possible with a function space approach. Another interesting perspective is to compare the predictive means with those obtain from NTK regression, which corresponds to the gradient-based learning with an infinitesimal learning rate.

The flexibility of DGP makes data fusion and multi-fidelity regression possible (Cutajar et al., 2019; Lu & Shafto, 2021a). The translation, which also includes log of the correlated prior over weights, then allows the neural network version of DGP multi-fidelity regression model. In such case, the regularizer contains the term $-\log p(\mathbf{w}_1 | \mathbf{z}, u)$ (see Sec. (4.2)), indicating the correlation between the components and the mode $\overline{\mathbf{w}}_1$ as a representation of low-fidelity data $\{\mathbf{z}, \mathbf{u}\}$ in weight space.

Lastly, the analysis of shallow trig nets in Sec. 3 suggests that the expressive power may be enhanced with i) adopting different weight prior distributions, which is equivalent to different GP kernels for function space regression, and ii) inserting phase networks before entering the sine/cosine activation units, which in principle removes the Gaussianity of the marginal prior distribution. Below, numeric simulations on real-world and toy data are present to support our findings.

### 7.1 Approaching the exact predictive mean with deep trig nets

Here, we are interested in predicting the trend of carbon dioxide concentration in Mauna Loa data set. It is well known that the GP regression with SE kernel fails to capture the short time scale variation as the prior density has its mass concentrated on smooth functions. We implemented using PyTorch the moment matching kernel correspond to the two-layer DGP with both kernels being SE (Lu et al., 2020) and the corresponding NTK derived in Sec. 5. The GP kernel regression with these two kernels (left: moment matching SE[SE] kernel, right: NTK) is shown in Fig. 1, in which the two results are only slightly different. The fixed form of kernels and the learned length scale $\ell_1 \ll 1$ in first layer leads to the constant predictive mean in the extrapolation. The fact that the rapid variation present in the training data is learned but not generalized can be considered a symptom of lack of feature learning.

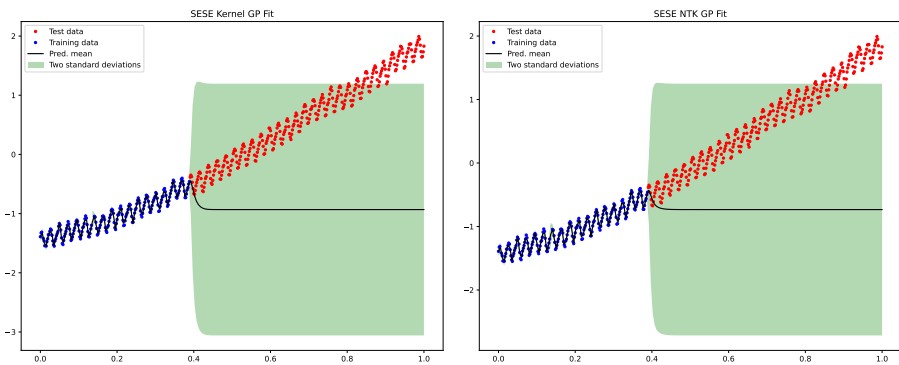

Figure 1: GP fitting the standardized carbon dioxide concentration data. **Left**: with the moment matching SE[SE] kernel. **Right**: with NTK.

With the translation from function space to weight space representation for DGP, it is interesting to apply the gradient-based learning for prediction. The two-layer DGP is then transformed into the two-layer trig network. We consider the squared loss together with the standard quadratic regularizer as the objective,

$$\mathcal{L} = \sum_i [y_i - f(\mathbf{x}_i)]^2 + \lambda \mathbf{W}^t \mathbf{W}. \tag{33}$$

Here, $\mathbf{W}$ stand for the flattened collection of weight parameters within the two layers, corresponding to the fact that the all the weights have independent and zero-mean Gaussian as prior. As for the random frequencies $\Omega_{1,2}$, they are samples from $\mathcal{N}(0, 1/\ell_1^2)$ and $\mathcal{N}(0, 1)$, respectively, and we kept them fixed in the process of gradient learning.

The two-layer trig network can have variation in the widths $n_{1,2}$ and the bottleneck width $H$, respectively. Fig. 2 shows the predictive means obtained with three variations in the network structure. Left panel displays the results from running with the six structures, namely $(n_1, H, n_2) = (2^{4:9}, 1, 300)$. Middle panel is for $(n_1, H, n_2) = (300, 1, 2^{4:9})$, and right panel is for $(n_1, H, n_2) = (300, 2^{0:7}, 300)$.

A few observations follow. The analysis in Sec. 6 suggests that deep trig network with the structure $(n_1 \to \infty, H = 1, n_2 < \infty)$ still converge to the limiting kernel $k_{\text{DGP}}$. This is in contrast to the structure $(n_1 < \infty, H = 1, n_2 \to \infty)$ leading to a deviation $\propto 1/n_1$ from the limiting kernel. Therefore, the inner width $n_1$ plays a more critical role in learning than $n_2$. In the left panel of Fig. 2, it is seen that when $n_1 \geq 64$ (green and above) the rapid variation in training data is learned. This *feature* is carried over to the future

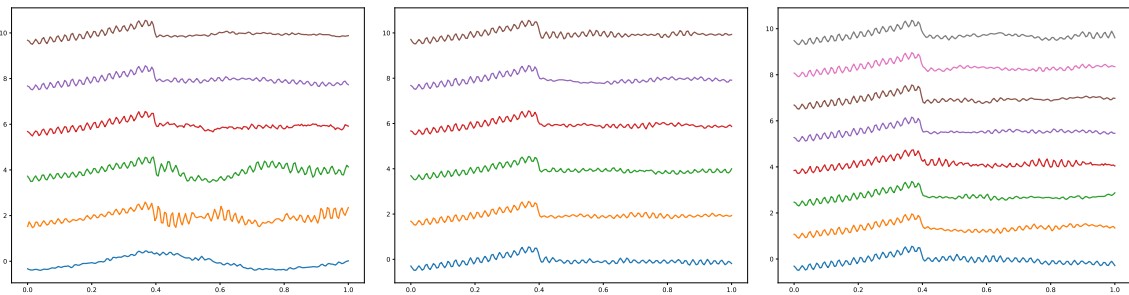

Figure 2: Predictive means from gradient-based learning with three sets of variation in network structures specified by the three widths $(n_1, H, n_2)$. We vertically shift these results for better visualization. Left panel displays the six results from $(n_1, H, n_2) = (2^{4:9}, 1, 300)$, middle panel for $(n_1, H, n_2) = (300, 1, 2^{4:9})$, and right panel for $(n_1, H, n_2) = (300, 2^{0:7}, 300)$.

times, but as $n_1$ increases the result is more close to that in Fig. 1. In the middle panel, the outer width $n_2$ does not seem to have effect on the learning and the generalization. Then, the variation in $H$ theoretically signifies the transition from DGP behavior to GP (Pleiss & Cunningham, 2021). In right panel, however, we do not see significant difference by varying the bottleneck width.

With the weight representation of the two-layer zero-mean DGP, we are able to approach the exact mean of intractable predictive distribution with the finite-width deep trig nets. Comparing with the kernel composition trick (Duvenaud et al., 2013) and the designed activation units (Pearce et al., 2020), we may conclude that, for this particular data, simply stacking two vanilla GPs into a DGP does not excel in enhancing the expressivity.

## 7.2 Toy multi-fidelity regression

DGP is a flexible prior exploiting the expressive power in compositionality, and an ideal model for fusing data from different levels of precision (Cutajar et al., 2019). Given the two-fidelity data $\{\mathbf{X}_1, \mathbf{y}_1\}$ (plentiful but low fidelity) and $\{\mathbf{X}_2, \mathbf{y}_2\}$ (rare but high fidelity), we may model the regression as inferring the composite function $f(x) = h(g(x))$ and the data are treated as observations, namely $\mathbf{y}_1 = g(\mathbf{x}_1)$ and $\mathbf{y}_2 = f(\mathbf{x}_1)$. It was shown in (Lu & Shafto, 2021a) that the moment matching kernel in Eq. (7) which takes the low fidelity data as the support for latent function $g(x)$ can reasonably well recover the truth function $f(x)$ even though the high-fidelity training data is rare. In the left panel of Fig. 3, we reproduced the simulation result in (Lu & Shafto, 2021a) with a PyTorch-based implementation.

In Bayesian learning, the structure of multi-fidelity DGP has the advantage of marginalizing the latent function $g$ conditioned on the low-fidelity data. As discussed in Sec. 4.2, the conditional mean and covariance for $g$ is translated into $\overline{\mathbf{w}}_1$ and precision matrix $A$ in weight space. Thus, the objective function for deep trig network learning becomes,

$$\mathcal{L} = \sum_i [y_i - f(\mathbf{x}_i)]^2 + \lambda_1 (\mathbf{w}_1 - \overline{\mathbf{w}}_1)^t A (\mathbf{w}_1 - \overline{\mathbf{w}}_1) + \lambda_2 \mathbf{w}_2^t \mathbf{w}_2 , \tag{34}$$

where the two regularizing terms come from minus log of the prior over weights. In the right panel of Fig. 3, one can see the predictive mean from using $\lambda_1 = 0.001$ (blue), 0.01 (orange), and 0.1 (green) given the high fidelity data (blue dots) generated from the true function (red dashed curve). As $\lambda_1$ increases, the knowledge, including uncertainty, about the latent function $g$ has more influence in learning the weight parameters through $\overline{\mathbf{w}}_1$ and $A$.

## 7.3 Expressive shallow trig nets

In the final subsection, we explore the possibility of enhancing the expressivity of shallow trigonometric net by i). sampling the random frequencies from a mixture of Gaussians with nonzero centers, and ii) in-

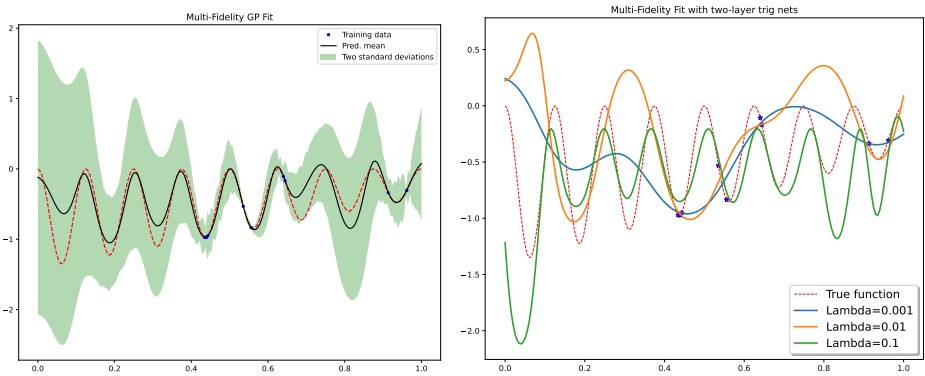

Figure 3: Network model fitting the multi-fidelity data. The aim is to learn composite function $f(x) = h(g(x))$ with plenty of low fidelity data (not shown) seen from $g(x)$ and very rare data seen from $f(x)$ (blue dots generated from the red dashed ground truth). Left: GP fitting with the moment matching kernel. Right: deep trig net fitting with varying regularizing strength $\lambda_1$'s.

serting a phase network before entering the sine/cosine activation units. With the shallow trigonometric net, we can apply the standard linear Bayesian learning if the random frequencies in the feature function $\Phi(\Omega\mathbf{x})$ are fixed. We generate three different sets of frequencies from different mixtures of Gaussians $\sum_i \mathcal{N}(\mu_i \sigma_i^2)$. We use $(\mu_i, \sigma_i^2, \#)$ to denote the component center, variance, and number of samples. In Fig. 4 one can see the predictive mean (black dashed) sandwiched by $\pm 2$ predictive std. The left panel is for a single Gaussian $\Omega \sim (0, 25, 75)$, middle for the mixture of $[(0, 5, 40), (50, 25, 35)]$, and the right for $[(0, 5, 25), (50, 25, 25), (100, 25, 25)]$. Given the same amount of activation units, the complexity of linear Bayesian model increases from the sampling $\Omega$ from a single zero-mean Gaussian to sampling from three Gaussians centered at 0, 50, and 100.

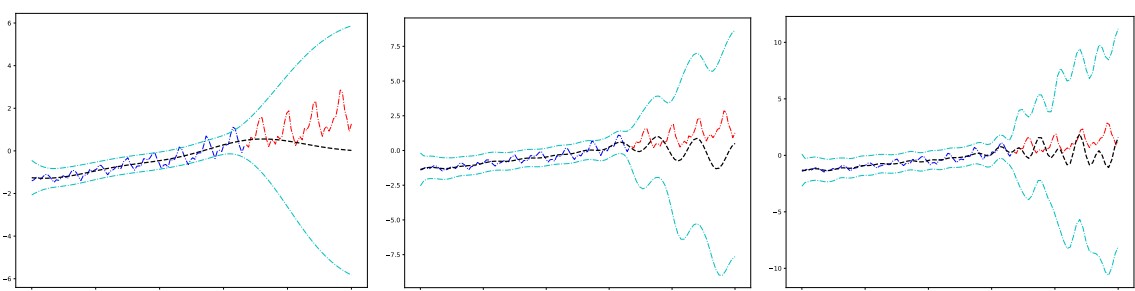

Figure 4: Bayesian linear regression on airline passengers data set with variations in selecting the random frequencies $\Omega$ discussed in Sec. 3. See text for details of the mixture of Gaussians.

Next, we are interested in fitting a pure noise data with the shallow trigonometric net. As discussed in Sec. 3.2, the shallow network in Eq. (16) with the inserted phase network $\Psi(\mathbf{x})$ is shown to have non-Gaussian marginal prior. To see if the non-Gaussian character is related to its expressivity, we consider four different setups for fitting the noise (red points shown in Fig. 5) generated from a normal distribution. In addition to the case without the phase network, a slight modification of Eq. (16) in changing the sign of $\Psi$ within the sine function will lead the marginal prior distribution back to Gaussian. We implement $\Psi$ with another shallow width-50 ReLu network using PyTorch. In Fig. 5, the predictive means from the vanilla GP and the shallow network without $\Psi$ are both linear with small slope, which is reasonable as the vanilla GP does not overfit. The phase network $\Psi$ does increase the expressivity of shallow network as the result (black solid) associated with Eq. (16) is more influenced by the outliers than the modified one (with $+/+$ sign for $\Psi$) is.

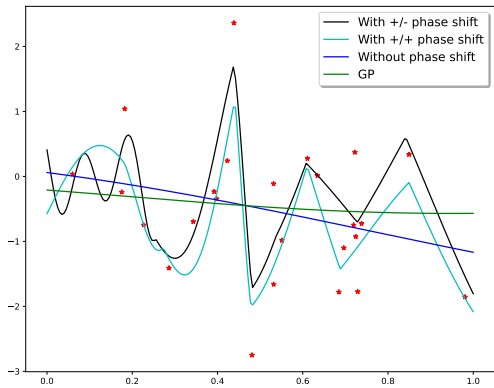

Figure 5: Fitting noise data (red dots) with GP and 3 shallow networks.

## 8    Related work

While Neal (1997) first pointed out the general correspondence between an infinite neural network and Gaussian process (GP), Williams (1997) demonstrated that neural networks with iid Gaussian weights and sigmoidal activation units are a representation of random functions drawn from a GP with arcsine covariance function. Later, Cho & Saul (2009) obtained the arccosine kernel from computing the covariance of outputs from the ReLu neural networks. Moreover, the correspondence holds beyond the shallow neural networks. Matthews et al. (2018) and Lee et al. (2018) studied the deep and wide neural networks and obtained a recursive relation for the emergent kernels. Similar techniques appeared in earlier work (Schoenholz et al., 2016; Poole et al., 2016) describing the statistics of forward and backward propagation with which phase transitions are identified in a number of learning phenomena. The connections between deep random networks and GPs were also studied extensively in (Yang, 2019), and detailed effects of finite width can be found in (Lee et al., 2020),

Theoretical progresses regarding understanding DGPs have been made via several important observations. In the deep limit, DGPs collapse to a constant function for some subspace of hyperparameters (Duvenaud et al., 2014; Dunlop et al., 2018; Tong & Choi, 2021) and carry a heavy-tailed distribution over function derivatives (Duvenaud et al., 2014). Lu et al. (2020) showed that the covariance and kurtosis are analytical characteristics of some two-layer DGPs, and a similar transition into chaotic phase with heavy-tailed multivariate statistics. Finite-width effects on statistics of the deep neural network were studied from field theory perspective (Antognini, 2019; Yaida, 2020; Roberts et al., 2021), NTK perspective (Hanin & Nica, 2019; Arora et al., 2019), and deep linear network (Aitchison, 2020).

Deep bottlenecked network representation of DGP in weight space was first proposed by (Cutajar et al., 2017), and (McDonald & Álvarez, 2021) generalized the idea to include the latent force model for composing the kernels. Agrawal et al. (2020) provided a formal and mathematical description for the connection. Uncertainty estimation in Bayesian deep neural network (Wilson & Izmailov, 2020) can be done with variational inference (Blundell et al., 2015), ensemble method (Lakshminarayanan et al., 2017), random dropout (Gal & Ghahramani, 2016), and Laplace approximation (Khan et al., 2019). The general issue about the under-estimated in-between uncertainty due to the independent weight assumption in approximate posterior was addressed in (Foong et al., 2020).

## 9    Conclusion

More precise understanding of deep learning is critical for exploiting its expressive power and potential applications in high-stakes domains. In the wide limit as well as the case with finite width, we analytically investigated the covariance, marginal distribution, and neural tangent kernel of the trigonometric networks, connecting them with the deep Gaussian processes which can carry squared exponential kernel,

spectral mixture kernel, and a combinations thereof. We have shown that deep Gaussian processes and deep trigonometric networks, one in function space and the other in weight space, yield the same covariance in a minimum model under various weight distributions. The derivation for the deep models in weight space is less intuitive, because it relies on an infinite dimensional Gaussian integral and knowledge of the spectrum of a particular random matrix. For deeper bottlenecked trig networks, the recursive relations [Eq. (25) in (Lu et al., 2020)] hold for the covariance approximately; without the bottlenecks the recursive relations [Eq. (22) in (Duvenaud et al., 2014)] can describe the covariance. We have open a door to analyzing the effect of the non-Gaussianity of deep Gaussian process on its modeling power. Specifically, the derived neural tangent kernel $k_{\mathrm{NTK}}$ with deep trigonometric net representation allows the possibility of analyzing the implication of differences between $k_{\mathrm{NTK}}$ and the exact kernel $k_{\mathrm{DGP}}$ of deep Gaussian process, and the data-dependent kernels as a result of finite-width.

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
