# OpenReview forum: "On Connecting Deep Trigonometric Networks with Deep Gaussian Processes: Covariance, Expressivity, and Neural Tangent Kernel"
_TMLR — Rejected by TMLR_

### Review · Reviewer_eudX · 2022-09-06

**Summary Of Contributions:**

While deep Gaussian processes (DGPs) are expected to offer greater modelling flexibility than their shallow counterparts due to their compositional structure, exact inference is often intractable, and one must resort to approximations for both training and generating predictions. In this work, the authors focus on the interpretation of DGPs put forward in Cutajar et al. (2017), whereby the random feature interpretation of covariance functions is leveraged to express DGPs in the weight-space view. In particular, the authors double-down on this connection in order to explore how trigonometric networks assimilate the results obtained by NTK regression, show how shallow GPs with the spectral mixture kernel can be replicated via trigonometric networks via sampling from mixtures of Gaussians, and also investigate the implications of using trigonometric networks with finite inner width. The contributions are primarily theoretical and shed further light on areas that were previously explored empirically. The paper concludes with a set of illustrative experiments showcasing the above contributions, including a toy example that illustrates how the trigonometric representation of a DGP can also be used for multi-fidelity learning.

**Broader Impact Concerns:**

I do not envisage any broader concerns related to this work that would require addressing.

**Requested Changes:**

While the contributions could be interesting to the community, I find the current submission to be far too sprawling and uneven to be ready for submission, and I would expect significant revisions to be applied in order to better convey the core ideas of the paper. In this regard, I believe that the paper should be quite majorly rewritten and restructured such that the contributions are more clearly expressed, and the overall implications of the findings become easier to follow.

I believe the Simulations/Experiments section should also be extended in a future revision - it is currently very limited, and does not demonstrate the expected thoroughness or completeness. It also doesn’t fit in well with the rest of the paper, and it wasn’t always easy to map each experiment to the corresponding contribution in the main text.

**Strengths And Weaknesses:**

- I appreciated the fact that the paper delves deeper into the theoretical aspects of deep Gaussian processes interpreted as deep trigonometric networks - there is always need for more papers which enhance the theory on existing models (in this case shallow and deep GPs using trigonometric features for approximation) and build connections with other emerging techniques (the Neural Tangent Kernel).

- While this submission draws elements from several prior works, the contributions are interesting and non-trivial, although the ‘pay-off’ itself is often unclear in the paper, and would benefit from being emphasised or highlighted more clearly. The broader implications of the findings could also be expressed more clearly in terms of the trade-offs on computational complexity and precision between different model formulations in light of the insights provided in this work.

- Unfortunately I found the paper very hard to follow most times. While an outline of the contributions is given in the Introduction, I later found myself struggling to understand where the discussions of related work end and the actual contributions begin. I believe this is also partly due to the diversity of contributions featured in this paper. On one hand, I understand and appreciate how the paper is intended to be a wide discussion of trigonometric networks and DGPs, but I personally think that the scope becomes confusing as additional concepts such as NTK regression and multi-fidelity modelling also get introduced. I would advise the authors to narrow down their focus or sequence them in a more incremental manner.

- Similarly, the simulations featured at the end of the paper come across as an afterthought, whereas they may have been much easier to understand if they were featured alongside the theoretical discussion earlier in the paper. The three experiments themselves are quite distinct and don’t build upon each other, which is why I believe they work better as an immediate follow-up to the proofs or discussion concerning the contribution, rather than a distinct section to conclude the paper. The experiments in general are very brief, and I would have expected a much more rigorous assessment of the various topics highlighted in the work. Instead, the experiments currently come across as very “one-shot”, with little evidence of how the results generalise across other datasets and set-ups.

- The paper contains a fair amount of typos or grammatical mistakes, and I would encourage the authors to look out for these in a future revision of the paper. I also wasn’t a fan of some of the shorthand that is sometimes used inconsistently within the paper (e.g. ‘trig’ instead of ‘trigonometric’).

---

> ### Author Response · Authors · 2022-10-11
> **Response**
>
> We thank you for your comments which nicely summarized our motivation and goal.
>
> $\bf Technicality$:
>
> We appreciate that the discussion of NTK and its relation to the mean of the intractable predictive distribution are kindly noted by you. However, partly inspired by the other two reviewers, we find that the bottleneck inside the deep trigonometric network actually breaks the assumption of an infinitely wide deep neural network. To enhance the clarity and remove concerns on technicality, it is best to reduce the claims related to NTK.
>
> Specifically, although the derivation in Lemma 7 remains correct, the entire Sec, 5 will be rewritten and be moved to a new section discussing future questions. In addition, the sentence “Interestingly, the network representation … reveal the mean of intractable predictive distribution” in Abstract, the last sentence “ More interestingly, the MAP … (NTK) regression” in the first paragraph in page 2, and the (iii) point in the third paragraph in page 2, will be removed.
>
> $\bf Clarification$:
>
> We will follow your suggestion of moving the numerical simulation in close proximity with the theoretical parts. In the new version, we will add new simulations on a high dimensional data set, e.g. MNIST, and compare it with other models.
>
> $\bf Responses\ to\ the\ requested\ changes$:
>
> We shall take the chance to rewrite the paper, reducing the claims about NTK, and adding more simulation to reflect the uncertainty estimate and other quantitative metric.

---

### Review · Reviewer_ojZV · 2022-09-09

**Summary Of Contributions:**

This paper investigates a parametric approximation to the MAP solution of Deep Gaussian Processes (DGP) - referred to as Deep  Trigonometric Networks - in which each GP layer is approximated with a finite Fourier basis. The resulting model is similar to a bottlenecked neural network. The authors make the following contributions:

- Compute the prior covariance of Shallow and Deep Trigonometric Networks (assuming a random distribution over weights) in the infinite and finite-case limits.
- Compute the NTK of Deep Trigonometric Networks

**Broader Impact Concerns:**

None, as far as I can tell.

**Requested Changes:**

## Critical changes

- Update the motivation of the paper. As it stands, if you are investigating a MAP approximation to DGP, you cannot claim that an advantage of DGP is "principled uncertainty estimation." Without this uncertainty estimation / the nonparametric nature of DGP, it is rather unclear what their advantage over standard DNN are.

- Fix all incorrect claims (see "Weaknesses" section).

- Many of the results are based on subjective qualitative assessments. (e.g. "we do not see significant difference by varying the bottleneck width"). What does significant mean? One dimensional predictive means and draws from the prior are hard to qualitatively analyze ([Stephenson et al., 2022](https://arxiv.org/abs/2106.06510)). It would be very useful to back up these qualitative assessment with some sort of quantitative metrics (the Stephenson paper has some good examples).


## Non-critical changes (but changes that would make the paper much stronger)

- The results section of this paper leaves more to be desired. A few thoughts on how it can be strengthened:
  - High dimensional datasets (or at least datasets with more than 1 dimension)
  - Comparison to other inference approaches.

- Add more experiments comparing the NTK versus the GP, and add a better argument for why it is interesting to investigate the NTK of Deep Trigonometric Networks. Alternatively, cut the NTK derivation from the paper.

- Figure 2 could use a better legend (label the different line colors)


## Typos (non-critical)

- "Spectral mixture" is often mis-spelled. For example, "spectra" on page 6, and "spectrum mixture" in Lemma 2.
- You are inconsistent with your capitalization of certain scalar parameters. For example, on page 3: you use $D$ and $d$ interchangeably to refer to the input dimensionality. Also, in the proof of Lemma 7, you use both $k_\text{DGP}$ and $K_\text{DGP}$ to refer to the DGP kernel function.

**Strengths And Weaknesses:**

## Strengths

- The derivations of this paper are mostly sound (except for a few; see below).
- The method analyzed in this paper provides a mathematically well-motivated approximation to the DGP posterior mode, with significantly simpler inference than other methods.

## Weaknesses

### Motivation and methods do not align

The authors begin the paper describing the strengths of DGP as follows:

> DGP..., along with principled uncertainty estimation inherited from Gaussian Processes (GP), aimed to exploit the compositional structure like DNN for superior expressivity and feature learning.

However, the authors also claim the following reasons for investigating Deep Trigonometric Network approximations of DGP:

> An important consequence of the translation between DGP in weight representation and function representation is that one can pursue the MAP estimate of weight parameters from the exact posterior. The point estimate then allows to obtain the mean of predictive prediction, which does not seem possible with a function space approach.

These two ideas seem very contradictory: on one hand, the motivation for using DGP is that we get principled uncertainty quantification. However, we lose that uncertainty quantification if we're doing MAP estimation. A Deep Trigonometric Network is basically the same thing as a neural network with only two minor changes: 1) we use the activation functions $\Phi(\Omega_i(\cdot))$ rather than a ReLU and 2) we don't apply a learned parameter matrix to the inputs.

To this end there doesn't really seem to be any advantage of MAP-estimated Deep Trigonometric Networks (ultimately what this paper claims is a straightforward way to perform DGP inference) and doing standard Deep Neural Network training with a modified activation function. I won't make any subjective claims about whether Deep Trigonometric Networks would be of interest to the community, but the authors should justify why these models substantively differ from neural networks if we are simply going to do MAP estimation.


### Clarity / lack of cohesive narrative

In many ways, the organization of Sections 4-6 makes this paper feel like a collection of mathematical derivations rather than a coherent narrative of results. For example, while the authors do demonstrate the limiting covariance of Deep Trigonometric Networks, derive its NTK, and demonstrate a finite-width correction, after reading the paper I am left unclear what to do with these results. Are these results a justification for using Deep Trigonometric Networks, or are they a way to better understand their properties? The experiments in Section 7 don't adequately show me what can be done with these results (see requested changes).

Again, I don't wish to make a subjective claim about whether or not these results will be "impactful," but I do think that there is an opportunity to better organize these results so that people within the DGP/DNN community know how to interpret/make use of these mathematical derivations.

There are also notational issues (see typos in requested changes), and some of the terminology could be made more precise.
- The use of "weights" to describe $w$ is a bit confusing,. as it could also be a reasonable term to define the fixed parameters $\omega$. You should at the very least make this terminology more clear.
- The concept of "hyperdata" is mentioned a lot before it is introduced, and it should be more thoroughly introduced since it is not a well-known concept.



### Some claimed contributions are trivial or well known

> Contribution 2: "we show shallow trig networks can approximate a GP with spectral mixture kernel... if the features are samples from a mixture of Gaussians"

With all due respect, I do not think that this can be claimed as a significant contribution. The authors note: "Following the work of Rahimi and Recht (2008), one can obtain a shallow trig network representation of GP regression model with SE kernel...". In fact, the argument of Rahimi and Recht (2008) argue is that we can obtain a finite basis approximation (or rather, a shallow trig network representation) of any kernel with a known spectral density. The spectral mixture kernel is explicitly defined by Wilson and Adams (2013) through its spectral density, which makes it trivial to construct a finite basis approximation/shallow trig network representation.

> Remark 3:

This is a minor note: but the proof that G has one non-zero eigenvalue can be simplified down to one line (the matrix is the outer product of one vector).


### What is the purpose of the NTK derivation?

The justification for deriving the NTK given in Section 1 is very vague, and possibly incorrect (see below). Moreover, the results section barely even makes use of the derived NTK: there is only a single experiment that is hard to interpret (the choice of model is not well suited to the data, and there is little discernible difference between the GP and the NTK.)

As it stands, the derivation of the NTK in Section 5 feels more like a mathematical exercise than an important finding. I would suggest that the authors better demonstrate (empirically) how this result could be used, or they should potentially consider cutting this section from the paper.


### Some claimed contributions appear to be false

> The expectation is that kernel regression using NTK shall correspond to the exact MAP solution of DGP.

I fail to see why this is expected. Firstly, the NTK solution only holds for models with very wide models (or, in the case of Deep Trigonometric Networks, models with very wide bottlenecks). Secondly, even if a neural network is very wide, it is well established that regression with the NTK does not correspond to the MAP solution of the corresponding Bayesian neural network (see e.g. [Lee et al., (2019)](https://arxiv.org/abs/1902.06720)). The MAP solution of a Bayesian neural network corresponds to kernel regression with the limiting GP kernel; not the NTK kernel. Since Deep Trigonometric Networks are a special case of standard neural networks, I don't understand why we would expect different behavior.

> Lemma 3

The authors claim that incorporating a phase shift network "lifts the Gaussianity" of a shallow trig network. I don't think that this is true, unless the shift $\psi(\mathbf x)$ is a random variable. (Proof: if $\phi(\mathbf x)$ is not a random variable, then $cos[\omega_i \cdot \mathbf x + \psi(\mathbf x)]$ and $sin[\omega_i \cdot \mathbf x + \psi(\mathbf x)]$ are fixed quantities w.r.t. $\mathbf x$. $f_\psi(\mathbf x)$ then corresponds to a linear combination of $w_i^c$ and $w_i^s$ variables (where the coefficients of the linear combination are deterministic w.r.t. $\mathbf x$), which is a Gaussian random variable.

If $\psi(\mathbf x)$ is assumed to be a random function, the authors should state so. No information about $\psi(\mathbf x)$ is given.


### Other notes

> [Section 4] ... which corresponds to the zero-mean two-layer DGP with SE kernels.

It corresponds to a two-layer DGP where 1) the first layer uses a SE kernel, but 2) the second layer uses the sum of one-dimensional SE kernels, where each of the SE kernels is applied to each of the $H$ functions output from the first layer. To the best of my knowledge, the more general (and more common) case of a two-layer DGP with non-additive kernels (e.g. Damianou and Lawrence (2013), Salimbeni et al., (2017)) cannot be constructed from a Deep Trigonometric Network. You should be careful to specify this in the paper.

> Closed form covariance functions can be derived for shallow networks... but the same techniques do not seem to cary to deeper networks.

This is not true. Lee et al. (2018) derived closed-form expressions for deep ReLU networks. Admittedly, these closed-form expressions are defined through recursive formula, but they sill qualify as "closed form" solutions nonetheless.

---

> ### Author Response · Authors · 2022-10-11
> **Response**
>
> We want to thank you for your very insightful and motivating comments.
>
> $\bf Technicality$:
>
>  - Lemma 3: The fact that the $\Omega$’s in Eq. 16 are random variables, not fixed numbers, may cause your misunderstanding. Hence, a fixed and nonzero Psi(x) could prevent the conditional variance for $f |\Omega,x$ from becoming a constant, and therefore the Gaussianity in the marginal prior is lifted.
>
>  - NTK: We thank you very much for your insight regarding NTK versus DNNGP in the case of the Bayesian deep neural network regression. The other reviewer raised similar concerns regarding NTK. We have to admit that the present paper does not have sufficient results, theory or experiment, to support the connection between NTK and the MAP of true predictive distribution. We will reduce the claim regarding NTK in the revised version. Besides, we do not wish to confuse the readers who expect the derived NTK for DGP may share the properties of the NTK for infinitely wide DNN, e.g. NTK being invariant during the gradient descent. These interesting issues deserve further investigation, and will be discussed in future works.
>
>  - Remark 3: You are correct. The reason for having only one nonzero eigenvalue of G is its outer product form.
>
>  - Sec. 4 and non-additive kernel in DGP: To the contrary, we think the second GP uses the non-additive kernel, i.e. the product kernel $k(f1(x), f1(y))=\prod k(f1_i(x),f1_i(y))$. The notion is more clear in the more recent paper (see Appendix G in Pleiss and Cunningham 2021) where you can see Eq. (21) is the same as that in Sec. G.2. So we shall mention in the revised version that the deep trig network can not represent the 2-layer DGP where the second GP uses additive kernel.
>
>  - Iterative expression for NNGP kernel is closed form?: We accept your notion of closed form, and in the revised version we shall remove the sentence “ .. but the same techniques do not seem to carry to the deeper networks”.
>
> $\bf Clarification$:
>
>  - The motivation in the absence of uncertainty estimation: Our perspective is that the true posterior over the latent functions when using DGP in Bayesian regression is not known, to our best knowledge. So if there is a way to partially know the true posterior, i.e. the MAP, it should be interesting to some people in the DGP community. Of course, to obtain the MAP is practically straightforward once the translation from function space to weight space is done. We believe that our paper provides the important piece in connecting DGP to the deep trig network: the covariance in the wide limit in weight space can approach that obtained in the function space.
> As for the uncertainty estimation, motivated by your suggestion, we have been working on implementing a simple and sample-based algorithm so the predictive uncertainty can be reported for data fit. We shall put it in the revised version.
>
>  - Triviality of Lemma 2: Yes, we agree that the two papers pointed out the essential grounds that lead to Lemma 2, but, to our best knowledge, we are not aware of any paper explicitly mentioning the idea of constructing a finite shallow network to approximate a GP with spectral mixture kernel. Moreover, we wish to show the freedom in choosing kernels in GP and DGP is not lost when the function-based models are translated to the weight-based ones. Lemma 2 is an easy example for understanding.
>
> $\bf Responses\ to\ the\ requested\ changes$:
>
>  - Our next action is to insert a section discussing a sample-based way to estimate uncertainty with the trigonometric networks.
>  - We shall reduce the claim of NTK on relating to the MAP of DGP posterior, and the implication of NTK for the bottlenecked deep network remains an open and interesting question. New investigation along this line will be the future work.
>  - Specifically, although the derivation in Lemma 7 remains correct, the entire Sec, 5 will be rewritten and be moved to a new section discussing future questions. In addition, the sentence “Interestingly, the network representation … reveal the mean of intractable predictive distribution” in Abstract, the last sentence “ More interestingly, the MAP … (NTK) regression” in the first paragraph in page 2, and the (iii) point in the third paragraph in page 2, will be removed.
>  - We will cite the Stephenson el. al. AISTATS paper.
>  - Extension of current regression setting to classification on high dimensional data sets, e.g. MNIST is under construction. The trigonometric network output will be sent to softmax to produce categorical probability.
>  - We thank you for your careful reading and very useful suggestions.

---

> > ### Comment · Reviewer_ojZV · 2022-10-14
> > **Response**
> >
> > Thank you for the clarifications and the response.
> >
> > > Sec. 4 and non-additive kernel in DGP: To the contrary, we think the second GP uses the non-additive kernel, i.e. the product kernel.
> >
> > My mistake. Thank you for clarifying.
> >
> > > We will cite the Stephenson el. al. AISTATS paper.
> >
> > To clarify: I was not asking you to cite the paper. I was pointing out that - based on the findings of the Stephenson et al. paper - qualitative assessments of GP samples are very misleading. This is part of a larger comment that the results section of the paper is wanting for more experiments (and preferably experiments with more rigorous quantitative assessments).
> >
> > > Our next action is to insert a section discussing a sample-based way to estimate uncertainty with the trigonometric networks.
> >
> > I'm pleased to hear that this is a next step, though this sounds like a very substantial new contribution.

---

### Review · Reviewer_mJRo · 2022-10-04

**Summary Of Contributions:**

The authors are concerned with showing that deep Gaussian processes and deep trigonometric networks can yield the same covariance function. The machinery is demonstrated on some toy experiments.

**Broader Impact Concerns:**

No concerns.

**Requested Changes:**

- The writing could be improved. Most of the time the problems are minor (E.g. there is awkward phrasing throughout, "Mean of the estimator only coincides with the exact..." instead of "The Mean of the estimator..." and the whole last paragraph of section 1. However sometimes the writing leads to misunderstanding the technical content. For example, in the first paragraph of section 2, it is not "the (singular) goal", but rather "a goal". Another example immediately follows, "Namely, XYZ becomes the predictive solution with...". What does "predictive solution" mean? The MAP is not enough to characterise the distribution, so how does the distribution become the MAP?
- Last sentence of 2.1. ", but the same techniques do not seem to carry to the deeper networks." Can you be more precise here? Which "same techniques"? There are a few works that describe the convergence to a GP for deep neural networks, and the covariance function may be evaluated using an iterative expression, as you point out in section 8. But the way this is written in 2.1, it is not obvious that this is the case.
- Section 2.2. "with which the optimal hyper-parameters is determined." Can you elaborate? I guess here you are referring to type II maximum likelihood, but there are many other possibilities too.
- Where does Lemma 3 come from? Is it in Yaida, 2020?

**Strengths And Weaknesses:**

My main concerns are about the clarity and correctness of the mathematical results. For example:
- Lemma 2. I don't see why an infinite width limit is needed. In the proof, the expectation on the left hand side is a deterministic object, which is equal (not \to) an integral. It can be computed without taking an infinite width limit. An infinite width limit would only be required if we looked at an inner product of (finitely many) activations. This actually holds for all shallow neural network type covariance functions, under mild conditions. The covariance function can be evaluated exactly, but the stochastic process is only Gaussian in the limit.
- I don't follow the proof of Remark 2. Under what conditions is this true? Is $p(w)$ required to be Gaussian? Even in this case, I can't see how it is true, for an arbitrary distribution $p(\Omega)$. Why is the first sentence of the proof true?
- Proof of lemma 4. What is $G$? Is this related to the $G$ appearing in the proof of the previous lemma?
- Proof of lemma 4. The first expectation in the second line of the proof (only over $\mathbf{w}_2$) does not need a limit, but the second one does. This is where maybe you hope to invoke Lemma 2 (?), which should be corrected to relate to an infinite width limit. Then the expectation is actually only required over $\mathbf{w}_2$, not all of the weights.
- First sentence of section 4.2. Why and in what sense is this true?
- Proof of lemma 7. "Where we observe that the first term is the same as the covariance of DGP." Why is this true?
- "Notice that the order of differentiation and expectation can be switched" --- Why?
- The definition (26) is motivated in the original NTK work via an appropriate limit of networks trained using gradient flow. Where does the motivation for (26) come from here?
- Text after Lemma 8. I can't follow this proof.  What does the subscript $\hat{k}_{SE}$ on the expectation mean? Which central limit theorem are you using? What does $\approx$ mean?

My second concern is the overall message of the paper. I have to admit that I did not understand what the paper was really about. The covariance function of neural networks with trigonometric activations is known, and can also be written down for the NTK. Is the idea here that deep Gaussian processes have the same covariance function? Why is this interesting, practically useful, or conceptually insightful? Is it possible to simplify the message to make it clear what the story of the paper is? I did not understand the relevance of the experiments.

# SUMMARY:
I cannot recommend that this paper be accepted, without (1) clarifying and making correct the mathematical statements and (2) clarifying the story of the paper. I do not think this will be possible without significant changes to the paper.

---

> ### Author Response · Authors · 2022-10-11
> **Response**
>
> We thank you for the feedback on our submitted work. Below are our responses.
>
> Technicality:
>
>  - Lemma 2: $\Omega$’s are the numbers sampled from the Gaussian mixture distribution, while the weight parameters are random variables. Thus, the expectation is applied only to the weight, not to the Omega’s. So a finite sum of cosine terms is spilled out of $E[~]$. With the infinite width limit, the integral represents the sum.
>
>  - Remark 2: Following the study of marginal prior over the linear network output (see Remark 1), it investigates the shallow trigonometric network with the approach used in Zavatone-Veth and Pehlevan 2021. Namely, what is the marginal distribution over $f(x)$ in Eq. 10? It should be helpful to first consider the trivial case f(x) = wx with w being a Gaussian RV with zero mean and variance $\sigma^2$. Then the distribution over $f(x)$ is the also a zero-mean Gaussian but with rescaled variance $(x\sigma)^2$. Then Remark 1 discusses the case for $f(x) = w\Omega x$ with both $w$ and $\Omega$ being Gaussian. Now we must integrate out the Omega (see details in Remark 1), and the resultant distribution is non-Gaussian. With this setting, Remark 2 discusses the case when f(x) is given by Eq. 10, and we see the effect of marginalizing the Omega on the resultant distribution: When conditional on the Omega, the conditional variance is $\sin^2(\Omega x) + \cos^2(\Omega x)$, which becomes constant, no longer depending on Omega. Thus this case is similar to the trivial case even though the extra Omega being involved.
> We apologize for  the confusion regarding the Omega’s here which are random variables in the finite width case, rather than the samples discussed in the previous bullet point.
>
>  - Lemma 4: Yes, G is defined in Eq. 20. Similar to our response in the first bullet point, the issue should be resolved by stating that the weight parameters w1 and w2 are random variables but the feature parameters $\Omega_1$ and $\Omega_2$ are samples in the infinite width limit.
>
>  - Lemma 7: (a) equality with $k_{DGP}$: the function $f(x)=\sum w_{2,i}h_i(x)$. The first term of your question is a dot product between two gradients, which generates $\sum h_i(x)h_i(y)$. It can be observed that the $E[f(x)f(y)]$ with respect to w2 in Eq. 21 spills out the same expression if one takes the variance of $w_2$ to be one. Thus, the first term in NTK is the same as $k_{DGP}$. (b) exchangeability: as long as the object under integration is continuously differentiable, the switch of order is legitimate. In this case, the object is $f(x)f(y)$ and it is continuously differentiable with respect to the weight $w$’s.
>
>  - Lemma 8: As the finite width effect is discussed in Sec. 6, the covariance function now is not fixed but depends on the realization of the $\Omega$’s (see Eq. 29). There, we wish to consider the expectation of the sample-dependent covariance function. It can be observed in Eq. 29 that the Omega1’s enter the covariance through the sum of n1 cosine terms, which we label as $\hat k_{SE}$. Therefore, instead of marginalizing each of the $\Omega_1$, we can consider the sum as a new RV, and it is clear that if $n_1$ is large, the sum can be approximated as a Gaussian based on the central limit theorem. Thus, the evaluation of expectation in Eq. 30 is not exact, but a reasonable approximation when $n_1$ is large.
>
> Clarification:
>
>  - Main plot of the paper: while the approximate representation of vanilla DGPs in terms of trigonometric deep networks (Cutajar et. al. 2017) intuitively exploited the kernel decomposition idea proposed in (Rahimi and Recht 2008), there is a missing link between the approximate form of DGP and the exact covariance for a two-layer DGP discussed in (Lu et. al. 2020). With the link being established for a few cases in this paper, the readers from the DGP community should feel more comfortable with the approximating deep trig network. Secondly, as the practitioners have the flexibility to choose the kernels inside a DGP, our investigation shows that choosing weight parameter distributions is equivalent to choosing kernels (see Lemma 2). The less well known situation is that we can also translate the multi-fidelity DGP into the deep trigonometric network (see Lemma 5). Last but not least is the finite width effect where we analytically investigate the deviation in terms of kernel, Eq. (31), and the inference, Fig. 2.
>
>  - First sentence of Sec. 4.2: In the context, this particular sentence is to restate Lemma 4. Specifically, in the large width limit, the two-layer deep trig network with w’s and Omega’s specified in the text has the same covariance as the two-layer DGP with squared exponential kernel.
>
> Due to the space limit, see the second part of response below

---

> ### Author Response · Authors · 2022-10-11
> **Second part**
>
>  - Motivation for Eq. (26): The original definition of NTK arises in the situation when the deep neural network is infinitely wide in each layer. However, the deep trig network in our paper has a bottleneck layer. Thus, among other nice properties of NTK, the fact that NTK does not change with time during the gradient-based learning does not seem to hold for the kernel in Eq. (27) where we simply generalize the definition to the deep network with bottleneck. While the kernel derived from Eq. (26) should not be called NTK in the first place (the bottleneck NTK might be appropriate), the mathematical properties like dynamics during gradient descent and the implication for the kernel ridge regression remain open and interesting.
>
>  - We plan to reduce the claim related to NTK in the revised paper. Specifically, although the derivation in Lemma 7 remains correct, the entire Sec, 5 will be rewritten and be moved to a new section discussing future questions. In addition, the sentence “Interestingly, the network representation … reveal the mean of intractable predictive distribution” in Abstract, the last sentence “ More interestingly, the MAP … (NTK) regression” in the first paragraph in page 2, and the (iii) point in the third paragraph in page 2, will be removed.
>
> $\bf Response\ to\ requested\ changes$:
>
>  - We will follow your suggestion when rewriting the paper and be careful on the wording. As for the comments on the MAP, we want to stress that the translation of the vanilla DGP into the deep trigonometric network allows us to apply gradient descent to obtain the prediction corresponding to the mean of exact predictive distribution of DGP inference. However, the uncertainty in the exact predictive distribution remains intractable.
>
>  - Yes, for a deep and infinitely wide network, the technique used in Lee et. al. 2018 can be applied to obtain the exact kernel in an iterative expression. As the other reviewer mentioned, an iterative expression is a closed form too. We will avoid the ambiguity by removing the corresponding sentences, “ …, but the same techniques do not seem to carry to the deeper networks.” in the first paragraph on page 4, in the new version.
>
>  - If there is no more hierarchy regarding placing further prior over the hyper-parameters, determining the hyer-parameters through the marginal likelihood is a very standard approach. Please see the standard textbook, 5.4.1 in Gaussian Processes for Machine Learning by Rasmussen and Williams, for more details.
>
>  - No, Lemma 3 is one of our contributions.

---

### Decision · Action_Editors · 2022-11-11

**Recommendation:** Reject

**Comment:**


While the paper studies the interesting topic of connecting deep gaussian processes to deep trigonometric networks which would be of interest to TMLR audience, reviewers found issues with clarity / motivation / main messaging and technical validity of the claims. Authors were in agreement with some issues raised by the reviewers. All reviewers agree that the submission is not publication ready and would require considerable work and another round of review.

The action editor encourages the authors to take reviewers' comments and suggestions and fix or strengthen support of the claim and improve the overall paper.



**Audience:**

The topic of the paper is definitely of interest to many of TMLR audience. DGPs are expected to offer great modeling flexibility. Also correspondence and connections between different ML methodology provides interesting theoretical insights. Thus the studied topic is definitely of interest to the community when all issues in the "Claims and Evidence" section are addressed.

- Reviewer `mJRo`: "has potential to be of interest to the community, in its current state it is not suitable for publication"


**Claims And Evidence:**


There are few critical common issues raised by the reviewers:

- First is the clarity of the submission. In multiple instances, clarity of the writing has confused the reviewers and led to misunderstandings.
Reviewer `eudX`: "Unfortunately I found the paper very hard to follow most times.", "the scope becomes confusing …", "fair amount of typos or grammatical mistakes"

- Also reviewers pointed out that messaging and motivation of the paper was unclear and misguided.

- One critical technical issue raised by the reviewers concerned with connection to NTK with networks with bottlenecks, which authors agreed with potential technical issues with the claim. The authors agreed they will reduce/remove claims with regards to NTK.
Thus in short, the submission needs to improve on clarity and requires fixing potentially invalid claims